# Environmental DNA monitoring of oncogenic viral shedding and genomic profiling of sea turtle fibropapillomatosis reveals unusual viral dynamics

Jessica A. Farrell [1,2,5], Kelsey Yetsko[1,5], Liam Whitmore[1,3], Jenny Whilde[1], Catherine B. Eastman[1], Devon Rollinson Ramia[1], Rachel Thomas[1], Paul Linser[1], Simon Creer [4], Brooke Burkhalter[1], Christine Schnitzler [1,2] & David J. Duffy [1,2,3,4✉]

Pathogen-induced cancers account for 15% of human tumors and are a growing concern for endangered wildlife. Fibropapillomatosis is an expanding virally and environmentally co-induced sea turtle tumor epizootic. Chelonid herpesvirus 5 (ChHV5) is implicated as a causative virus, but its transmission method and specific role in oncogenesis and progression is unclear. We applied environmental (e)DNA-based viral monitoring to assess viral shedding as a direct means of transmission, and the relationship between tumor burden, surgical resection and ChHV5 shedding. To elucidate the abundance and transcriptional status of ChHV5 across early, established, regrowth and internal tumors we conducted genomics and transcriptomics. We determined that ChHV5 is shed into the water column, representing a likely transmission route, and revealed novel temporal shedding dynamics and tumor burden correlations. ChHV5 was more abundant in the water column than in marine leeches. We also revealed that ChHV5 is latent in fibropapillomatosis, including early stage, regrowth and internal tumors; higher viral transcription is not indicative of poor patient outcome, and high ChHV5 loads predominantly arise from latent virus. These results expand our knowledge of the cellular and shedding dynamics of ChHV5 and can provide insights into temporal transmission dynamics and viral oncogenesis not readily investigable in tumors of terrestrial species.

[1] The Whitney Laboratory for Marine Bioscience and Sea Turtle Hospital, University of Florida, St. Augustine, FL, USA. [2] Department of Biology, University of Florida, Gainesville, FL, USA. [3] Department of Biological Sciences, School of Natural Sciences, Faculty of Science and Engineering, University of Limerick, Limerick, Ireland. [4] Molecular Ecology and Fisheries Genetics Laboratory, School of Biological Sciences, Bangor University, Bangor, Gwynedd, UK. [5]These authors contributed equally: Jessica A. Farrell, Kelsey Yetsko. ✉email: duffy@whitney.ufl.edu

Sea turtle fibropapillomatosis (FP) is an epizootic (animal epidemic) tumor disease, affecting endangered sea turtles worldwide[1–5]. The disease is characterized by the formation of cutaneous and internal fibro-epithelial tumors, which can lead to debilitation and death. Fibropapillomatosis of sea turtles continues to spread geographically, and is now present in every major ocean basin in which green sea turtles (Chelonia mydas) are endemic (www.cabi.org/isc/datasheet/82638)[3,6–16]. Rates of FP are also increasing in many long-term affected locations, with incidence of the disease being the highest in near-shore habitats[3,9,17–21]. The FP-afflicted turtles studied here were found stranded in northeastern Florida. Fibropapillomatosis only began affecting C. mydas in northern Florida in the last decade, despite being present in areas of southern Florida since at least the early 1900s[5,12,19,22–24]. Concurrent with geographic expansion of the disease, incidence of FP in stranded C. mydas across the state of Florida has risen from 13.3% in 2005 to 42% in 2016[5,9,17,19].

Of all sea turtle species and life-stages, juvenile green sea turtles are most severely afflicted by FP. Long-lived reptiles have normally robust anti-cancer defenses, and with the exception of FP, reports of neoplasia in sea turtles are rare[25–29]. However, near-shore environmental exposures likely impair tumor suppressor mechanisms in sea turtles, thereby enabling virally-induced tumorigenesis[3,30]. While some environmental co-factors, such as eutrophication have been studied, their role in the pathogenesis of FP remains experimentally unconfirmed. It is possible that environmental exposures induce immunosuppression in sea turtles, thereby enabling viral loads to increase to the point of crossing an oncogenic threshold, similar to a number of human virally-induced cancers[3,5,31–33]. However, the pathogenesis of FP remains elusive. A chelonian-specific alphaherpesvirus (chelonid herpesvirus 5, ChHV5) has been implicated as a cause. However, Koch's postulates to confirm its causative role have yet to be fulfilled, because ChHV5 is extremely difficult to isolate and propagate in the laboratory[3,5,34–37]. Similarly, despite advances in FP tumor research[1,3,5,30,31,38–46], many open questions remain regarding the role of ChHV5 in driving FP tumorigenesis, including whether it is a cause of the disease or an opportunistic pathogen, exploiting immunocompromised tumor-afflicted turtles[3,5,31]. Interestingly, levels of ChHV5 (ChHV5 gB and UL30 gene DNA detected by qPCR, and ChHV5 glycoprotein H peptides detected by ELISA) in clinically healthy turtles are closer to that of FP tumors, than in non-tumored tissue of FP-afflicted turtles[34,47]. It is also unclear whether ChHV5 is lytic or latent in FP tumors and the occurrence of viral shedding and transmission is not well understood[3,5,48–51].

In many locations, such as Florida, live sea turtles with FP that are found stranded are admitted to rehabilitation facilities for treatment, which often includes tumor resection surgery. Sea turtles under treatment for FP provide a valuable opportunity to study this disease in a manner that is infeasible or impractical in free-ranging animals. The combination of accessible rehabilitating patients and modern transcriptomics and genomics of different tumor types can address the open questions relating to ChHV5's oncogenic role, while novel environmental DNA (eDNA) approaches can help resolve the dynamics of viral shedding and transmission.

Environmental DNA is a non-invasive forensics approach to the extraction and identification of organismal DNA fragments (genetic material) released into the environment, and this rapidly advancing approach is capable of improving endangered species detection and early pathogen detection[52–61]. Environmental samples can be analyzed for micro- and macro-organisms by several eDNA methods including metabarcoding and species-specific quantitative PCR (qPCR)[55,57,62]. The development of a rapid and high-throughput sampling scheme to detect virus shedding into the marine environment would benefit pathogen surveillance efforts, and consequently the performance of wildlife health status monitoring could be improved[48,63]. Here we applied qPCR and shotgun sequencing ("unbiased", non-barcoded) eDNA approaches to temporally quantify ChHV5 viral shedding from rehabilitating patients. Such novel approaches are a particularly beneficial feature of aquatic models of virally-induced tumors, and will enable greatly improved understanding of the dynamic relationship between viral load and viral shedding, which is not readily measurable in terrestrial species.

We demonstrated previously that ChHV5 was transcriptionally latent in a small cohort of seven established external FP tumors[5]. We postulated that ChHV5 might be latent in established tumors, but more active during crucial early stage tumor initiation events, akin to the 'hit and run' hypothesis of viral oncogenesis[5]. To investigate this hypothesis, we employed deep sequencing-based transcriptomics and whole-genome sequencing (WGS) to determine the viral load and transcriptional status of ChHV5 across a variety of FP tumor stages and presentations: new external, established external, post-surgical regrowth external, internal lung, internal urinary bladder, and internal kidney tumors. Analysis of the host aspects of the current study's transcriptomic and genomic data are explored in a companion paper[38]. Here we investigate viral dynamics during FP tumor growth, post-surgical recurrence, and correlations between ChHV5 viral load and ChHV5 gene expression with patient rehabilitation outcome to identify characteristics relevant to disease severity and fate of turtles afflicted with FP. Furthermore, we investigated potential routes of transmission, primarily horizontal transmission by direct shedding of ChHV5 into the water column, but also confirming the potential role of intermediary vectors and raising the possibility of vertical transmission. Fibropapillomatosis genomics[29,31] and eDNA-based pathogen monitoring can reveal the precise mechanisms through which the virus is transmitted and the role of ChHV5 in host cell transformation and tumor progression. Such research will provide insights into this wildlife epizootic and reveal how ChHV5 can rapidly induce novel cancer incidence on an epidemic scale. Such information is vital to enable improved management, treatment, and mitigation strategies to be developed to combat this sea turtle conservation-relevant disease epizootic.

## Results

**High prevalence of ChHV5 in marine leeches feeding on FP-afflicted turtles.** A number of potential routes for ChHV5 transmission between sea turtles have been postulated, including via vectors such as Ozobranchus leeches[51]. These marine leeches are commonly found on FP-afflicted turtles, often at high density within the crevices of external FP tumors (Fig. 1a, b). In line with previous studies[51], we confirmed that ChHV5 could specifically be detected from DNA extracted from leeches that had fed on FP-afflicted turtles (Fig. 1c–e). All leech pooled samples removed from FP tumors tested positive for ChHV5 DNA (Fig. 1c, d), while half of the leeches removed from non-tumor locations of FP-afflicted animals were ChHV5 positive (Fig. 1c). Of 30 leeches from FP tumors assessed individually (one DNA extraction per whole leech), 90% were positive for ChHV5 (Fig. 1e). The 3 leeches that tested negative for ChHV5 (Fig. 1e) had smaller (barely visible) blood pellets than the other 27 leeches. Leeches removed from FP-free animals did not test positive for ChHV5 (Fig. 1c).

**Detection of ChHV5 shedding into the water column by environmental DNA (eDNA) approaches reveals novel**

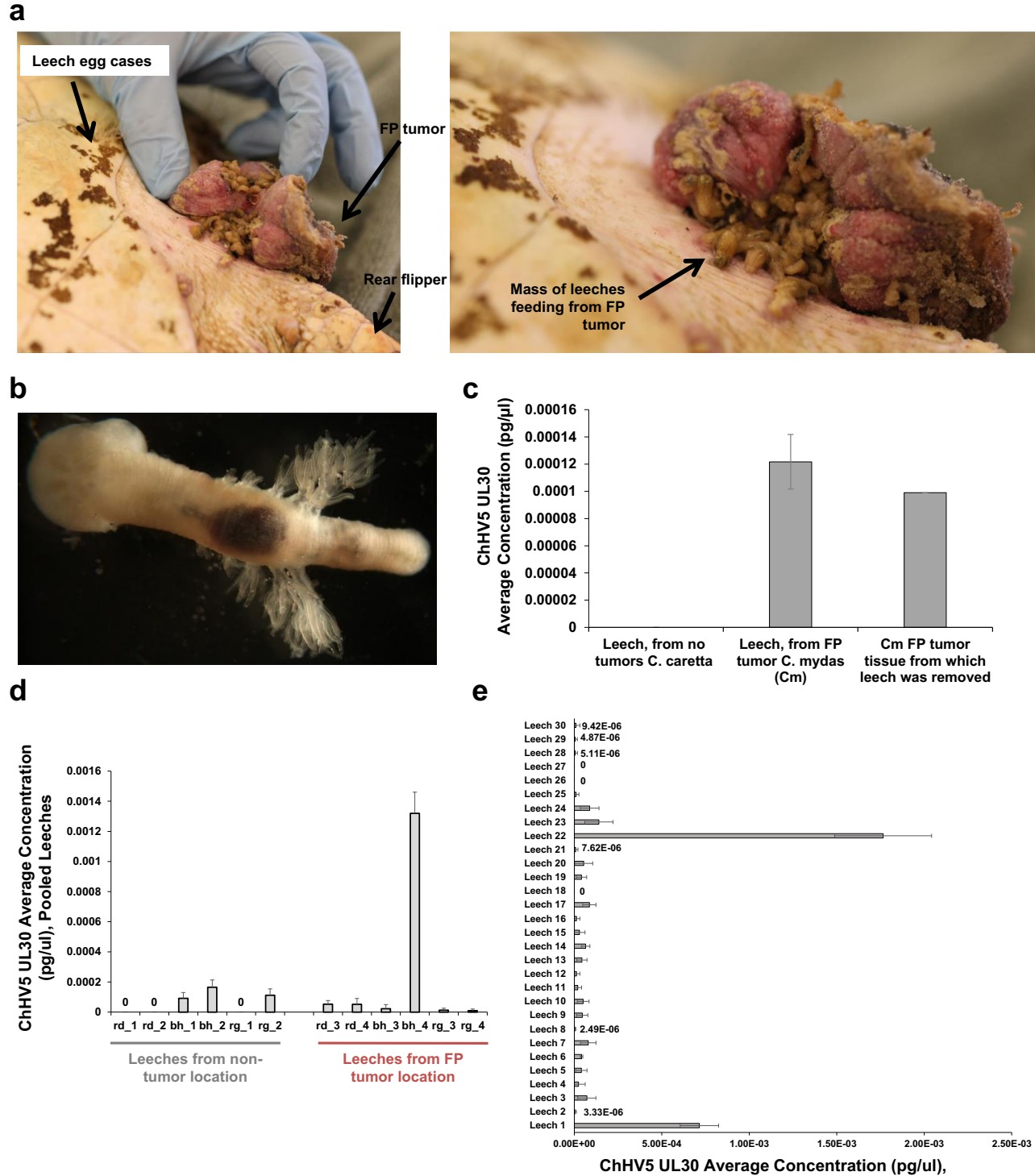

**shedding dynamics and tumor burden correlations**. Another potential mode of transmission is direct ChHV5 shedding into the environment. Direct viral shedding as a route of transmission is thus far only supported by indirect evidence, i.e., cloacal swabs, urine, ocular, oral, and nasal secretions[49,50,64], viral inclusion bodies near the surface of external tumors[48] and the elevated levels of ChHV5 detected in bladder tissue (see below). Direct detection of ChHV5 in the water column is lacking. Therefore, we employed environmental DNA (eDNA)-based approaches coupled with qPCR and next-generation sequencing to detect ChHV5 in patient tank water. Such novel approaches can aid in answering previously intractable questions about direct ChHV5 transmission, such as presence, abundance, and persistence in the marine environment. First, we determined that the presence of ChHV5 could be readily detected in eDNA extracted from tank sea water, using either an established[37] *UL30* qPCR assay (Fig. 2a) or next-generation sequencing (see WGS section below). ChHV5 was detectible not only from patient tank water, but also from sand after a turtle lay on it for 30 min while awaiting treatment procedures (Fig. 2b).

Importantly, not only was ChHV5 detectible, it was quantifiable, allowing comparisons between individual tanks and across

**Fig. 1 Leech ChHV5 detection. a** Green sea turtle inguinal external FP tumor, infested with leeches. Upon patient intake FP-afflicted tumors frequently harbor marine leeches, as was the case for this patient 02-2021-Cm "Broccoli". Leeches from "Broccoli's" tumors were used for the ChHV5 analysis in (**e**). Leeches are commonly found within the crevasses of external FP tumors (right image). **b** Detailed view of a marine leech removed from the surface of a fibropapillomatosis tumor, with gills and with dark red blood pellet (after feeding on a *C. mydas* turtle) visible. **c** Detection and quantification of ChHV5 *UL30* gene DNA by qPCR, using leeches as proxy eDNA samples (whole leech lysis and DNA extraction). Error bars denote the standard deviation of three technical replicates. Amplification ratio for leeches from the FP-free loggerhead turtle (09-2015-Cc) was 0, and the amplification ratio for both the FP-tumor leech and FP-tumor tissue samples (green turtle, 07-2015-Cm) was 1.0. **d** Quantification of ChHV5 *UL30* gene DNA by qPCR, from leeches removed from FP-afflicted green turtles from either FP tumors or non-tumor locations. Individual turtle denoted by rd—36-2020-Cm "Richard Dawkins", bh—52-2020-Cm "Bruno Hofer", or rg—78-2020-Cm "Ruth Gates". Approximately ten leeches were pooled for each of the 12 DNA extraction samples. Error bars denote the standard deviation of six technical replicates. **e** Quantification of ChHV5 *UL30* gene DNA by qPCR, from leeches (individual leech DNA extractions) removed from FP tumors of green sea turtle patient 02-2021-Cm "Broccoli" (**a**). Amplification ratios for leech samples are provided in Supplementary Table 1. Error bars denote the standard deviation of six technical replicates.

time (Fig. 2a–f). The level of detectible virus in patient tank water was positively correlated to the tumor burden of the patient(s) housed in that tank (Pearson correlation coefficient test, $R^2 = 0.5431$, $p = 0.0002$, df = 19) (Fig. 2a). Patients with large well-established tumors shed more virus into tank water than those with small new-growth tumors. Larger tumors shed more ChHV5 by virtue of their cumulative size, not because they contain more lytic ChHV5 per unit area than new growth tumors (see transcriptomics section below). As FP tumors were surgically removed, the level of ChHV5 in patient tanks dropped (Fig. 2c–e), suggesting that the tumors are the primary source of environmental ChHV5 (either through direct tumor shedding, or migration of virus throughout the body and excretion in bodily fluids). The level of ChHV5 in tank water of patients with high tumor burdens was higher than that seen in leeches feeding on FP tumors (Figs. 1c–e, 2a,c–e). Additionally, the quantity of ChHV5 eDNA in tank water was positively correlated with the *C. mydas* (green turtle) eDNA level (Pearson correlation coefficient test, $R^2 = 0.66$, $p = 0.00001$, df = 19), as detected by a custom *C. mydas* 16S rRNA DNA assay (Fig. 2f). ChHV5 eDNA was also detectible in the saltwater fishpond (approximate size 661,000 liters, Supplementary Fig. 1a) which receives the filtered outflow sea water from our patient tanks (Supplementary Table 1), highlighting ChHV5's potential persistence in aquatic environments.

**ChHV5 is latent in FP tumors, including early stage, regrowth, and internal tumors.** While further investigation of the genomic and environmental drivers of FP, and ChHV5 transmission is warranted[31], the suspected causal relationship between ChHV5 and FP also requires further study. Across all our sequenced FP samples (RNA-seq), ChHV5 transcripts were low, and we detected no major switch to active (lytic) virus in either new growth FP or post-surgical regrowth FP (Fig. 3a). In fact, levels of ChHV5 transcripts were only marginally (and not significantly) higher in FP tumors than they were in non-tumor tissue controls (Fig. 3a). The only significant difference found in the level of viral RNA transcripts between any of the groups was between regrowth and established growth external tumors (Kruskal–Wallis with Dunn–Bonferroni post hoc, $p = 0.013$). Furthermore, we detected no switch to active (lytic) viral transcription in internal tumors (Fig. 3a). Supporting the paucity of lytically replicating virus, no inclusion bodies were detected in either internal or external tumors (Supplementary Table 2) by hematoxylin and eosin staining (Fig. 3b). Together this suggests that the role of lytic ChHV5 in FP is minimal and that if ChHV5 is contributing to FP oncogenesis, it is either (i) occurring transiently, during very early tumorigenesis (before visible tumor appears), or (ii) through the expression of ChHV5 miRNAs (not assessed here), or (iii) that it is the latently expressed ChHV5 transcripts that are driving oncogenesis. We therefore next

examined the individual gene level transcripts to determine which ChHV5 genes were transcriptionally active in FP tumors (albeit at relatively low levels). Samples were grouped into four types: non-tumor, external FP (including established, new growth, and regrowth), kidney FP, and lung FP (Supplementary Data 1). Across all four sample types quite a consistent pattern of ChHV5 gene expression emerged (Fig. 3c, Supplementary Fig. 1b), with only 22 of ChHV5's 104 genes showing levels of expression above 1 transcript per kilobase million (TPM, Supplementary Data 2). Latency-associated genes, such as F-LANA formed part of this 22 gene group, being consistently expressed across all tumor types (Fig. 3c and Supplementary Fig. 1b). Given the paucity of lytic ChHV5, these 22 ChHV5 genes represent the most likely viral drivers able to contribute to FP initiation and ongoing tumor development and growth; therefore they warrant further functional investigation.

**Higher ChHV5 viral transcription is not indicative of poor outcome.** Almost all samples with ChHV5 viral reads above 200 per 10 million total reads originated from just three patients (Supplementary Fig. 2a). Interestingly all three of these higher viral transcript patients were successfully rehabilitated and released. We, therefore, investigated whether there was any relationship between number of viral transcripts and rehabilitation outcome. Counter-intuitively, patients with positive outcome (survived and released) on average had samples with statistically significantly higher ChHV5 transcripts (Mann–Whitney *U* Test, $p = 0.03$), while those patients that died in care or were euthanized due to advanced disease actually had lower ChHV5 transcripts (Fig. 4a). Even when internal tissue samples were removed from the analysis (as all internal samples originated only from the deceased/euthanized category), there remained a significant viral expression difference between the two outcome groups (Mann–Whitney *U* Test, $p = 0.0143$).

We next assessed whether individual viral genes correlated to patient outcome. Twenty-one genes were significantly differentially expressed between tumors from poor outcome (died or euthanized) and good outcome patients, all of which were upregulated in poor outcome tumors (Table 1). However, on closer examination of the poor outcome tumors, it was predominantly viral expression in internal tumors rather than external tumors that was responsible for this differential expression (Table 1 and Fig. 4b, c). Interestingly, the four ChHV5 genes which are atypical of alphaherpesvirus, F-lec1, F-lec2, F-sial, and F-M04[65], were more highly expressed in internal tumors (Table 1 and Supplementary Fig. 2b). These four genes are postulated to play a role in viral immune evasion and viral pathogenesis[65]. Of all of the differentially expressed ChHV5 genes, both genomic copies of the latency-associated F-LANA gene showed the largest fold increase in expression in internal tumors (Table 1 and Fig. 4b).

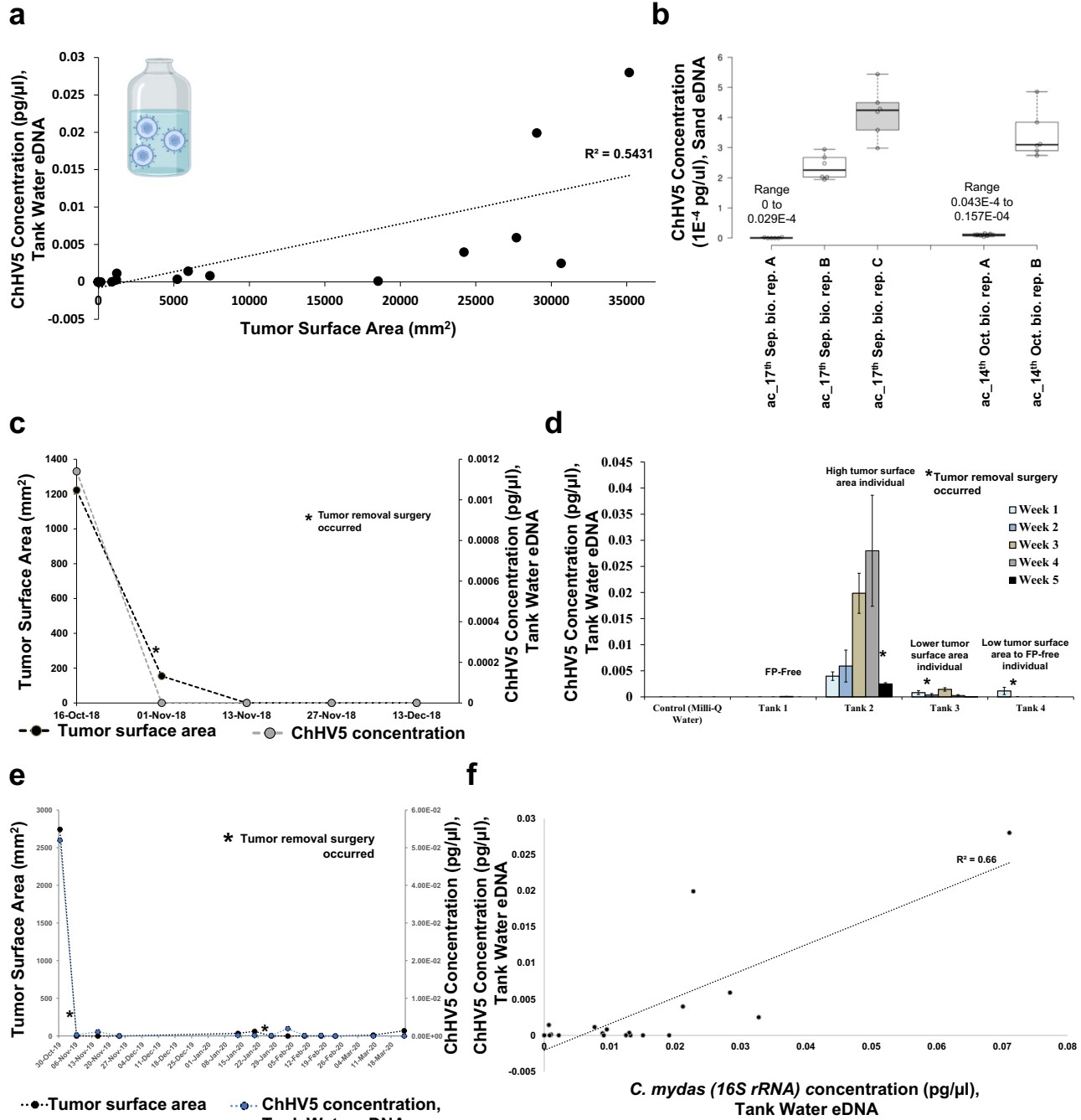

**Fig. 2 Environmental DNA (eDNA)-based detection, quantification, and monitoring of ChHV5 viral shedding into patient tank water, and sand. a** Correlation of individual patient tumor surface area (mm$^2$) and the concentration of ChHV5 virus being shed into their tank water, as detected by ChHV5 *UL30* gene DNA qPCR (positive correlation, Pearson correlation coefficient test, $R^2 = 0.5431$, $p = 0.0002$, df = 19). Water containing virus schematic insert was generated using BioRender (https://biorender.com/). **b** Detection of ChHV5 viral shedding onto sand which a patient ("Archie Carr" 49-2020-Cm) temporarily lay on while awaiting treatment. Detection of ChHV5 *UL30* gene DNA by qPCR. Amplification ratios for each sand sample are provided in Supplementary Table 1. Error bars denote the standard deviation of six technical replicates. **c** Patient time-course of tumor surface area changes (surgical removal) and concentration of ChHV5 shed into tank water, as detected by *UL30* qPCR. **d** Time-course of ChHV5 viral shedding into four patient tanks, as detected by *UL30* qPCR. Tumor removal surgery events are denoted by an asterisk. Note: the ChHV5 eDNA detected in tank 1 in week 4 was due to a second patient (FP-positive) being added to that tank for a single week, due to the rehabilitation needs of the hospital. Error bars denote the standard deviation of three biological samplings, each with three qPCR technical replicates. **e** Prolonged patient time-course of tumor surface area changes (surgical removal) and concentration of ChHV5 shed into tank water, as detected by *UL30* qPCR. **f** Correlation of *Chelonia mydas* (green sea turtle) eDNA shedding and ChHV5 eDNA shedding (positive correlation, Pearson correlation coefficient test, $R^2 = 0.66$, $p = 0.00001$, df = 19), both *C. mydas* (16S rRNA gene assay) and ChHV5 (*UL30* gene assay) eDNA were detected by qPCR.

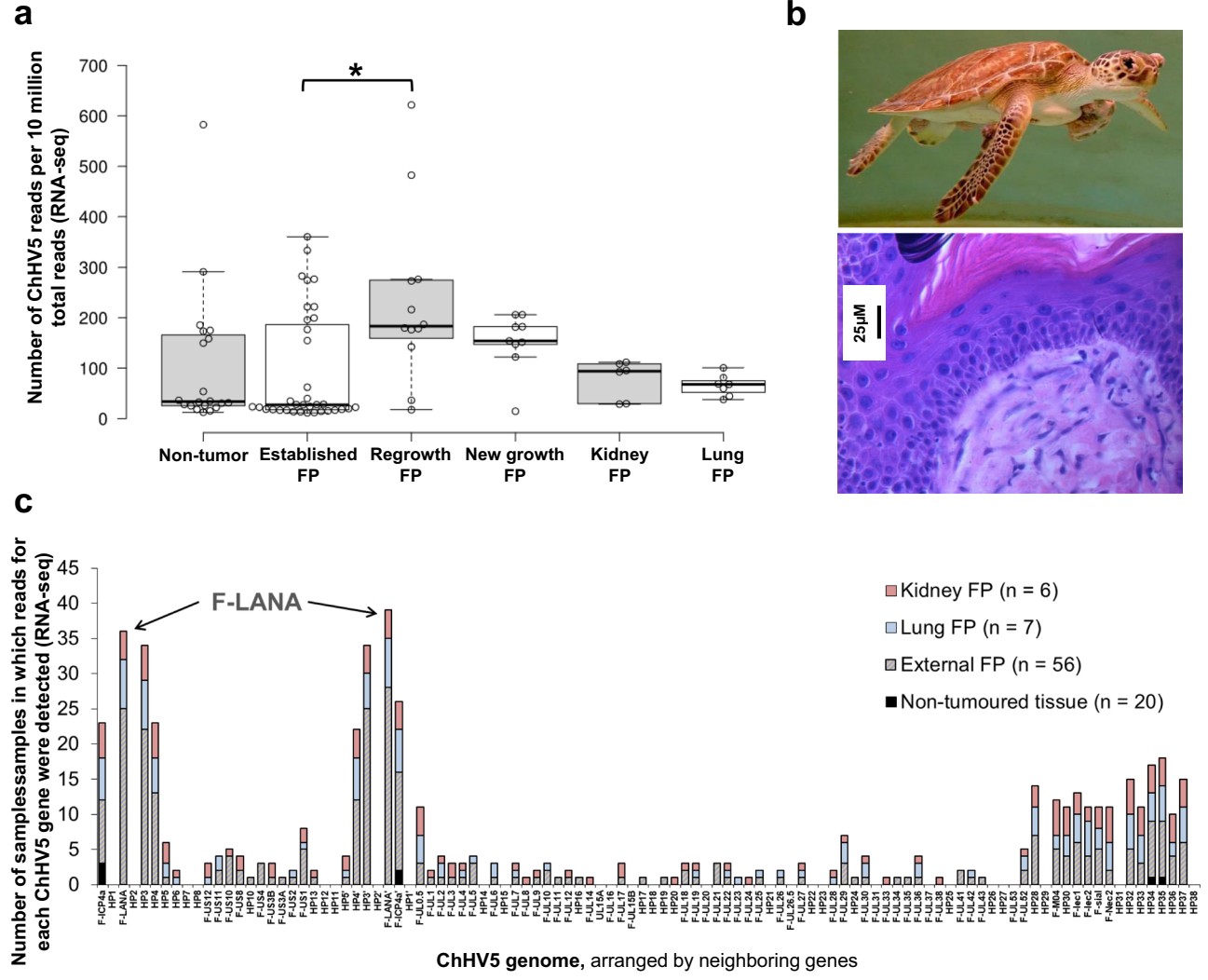

**Fig. 3 Transcriptomics of ChHV5 in external and internal FP tumors, and non-tumor tissue samples. a** ChHV5 expression across each sample type. Box plot with Tukey whiskers of the number of ChHV5 reads (RNA-seq) per 10 million total reads per sample. Individual sample values within each sample type are shown by the open points. Significant differences in averages between sample types were determined by a Kruskal–Wallis with Dunn–Bonferroni post hoc test and are denoted by an asterisk (*). Per sample type: non-tumor $n = 20$, established external tumor $n = 35$, regrowth external tumor $n = 12$, new growth external tumor $n = 9$, kidney tumor $n = 6$, lung tumor $n = 7$. **b** Top: Rehabilitating FP-afflicted juvenile green sea turtle. Image credit: Nancy Condron. Bottom: Hematoxylin and eosin stained fibropapillomatosis tumor, external regrowth tumor. **c** Total number of samples in which reads (RNA-Seq) for each ChHV5 gene were detected. A gene was counted as detected if a sample had TPM-normalized counts >0 for said ChHV5 gene.

**High ChHV5 viral loads predominantly arise from latent virus.** Given the consistently low level of ChHV5 transcripts in FP tumors, we next used whole-genome sequencing (WGS/DNA-seq) to quantify the viral load of ChHV5 to determine whether the low number of transcripts arises due to a lack of virus, or whether large quantities of virus are present within FP tumors with the majority of these being latent (not undergoing active viral replication and transcription). By not conducting any viral enrichment steps, the resulting read numbers give a more reliable indication of the relative abundance of viral DNA compared with host DNA (all within a single sample/library). Viral DNA sequencing reads in the FP tumors covered a broad range (Fig. 5a). Interestingly, the tank water eDNA had a higher viral load than plasma and non-tumor tissue from FP-afflicted turtles (Fig. 5a and Supplementary Fig. 2c). The lung and kidney tumor samples from the same patient (patient 27-2017-Cm) had dramatically different viral loads (Fig. 5a, Table 2). The six tumors (external and kidney) from patient "Yucca" (49-2019-Cm) had a range of ChHV5 reads per 10 million total reads (RPTM) from

1198 to 3127 RPTM. The new growth FP tumor from the patient "Lilac" (25-2018-Cm) had an intermediate viral load of 1,036 RPTM (Table 2 and Fig. 5a). Other new growth tumors from "Lilac" also showed high viral loads (Fig. 5b). Despite the high level of ChHV5 DNA sequencing reads in the lung tumor (3673 RPTM), this same tumor only had a very low level of ChHV5 RNA sequencing reads (68 RPTM), more closely resembling the read numbers of the non-tumor tissues (Table 2).

The kidney tumor from patient 27-2017-Cm had minimal viral load (Table 2 and Fig. 5a) and this same tumor had minimal copy number variations in the host tumor genome[30], suggesting that the tumor must be driven by other oncogenic mechanisms, such as point mutations, epigenetic changes, or transcriptional/translational aberrations. The viral load of this kidney tumor was within the range of non-tumor tissues (juveniles and hatchlings) and plasma from FP-afflicted turtles (Fig. 5a).

When DNA-based viral reads (Fig. 5a) were compared with viral reads from the RNA transcripts (Fig. 3a) of the 90 RNA-seq samples, the highest viral transcript load observed was ~600 RPTM

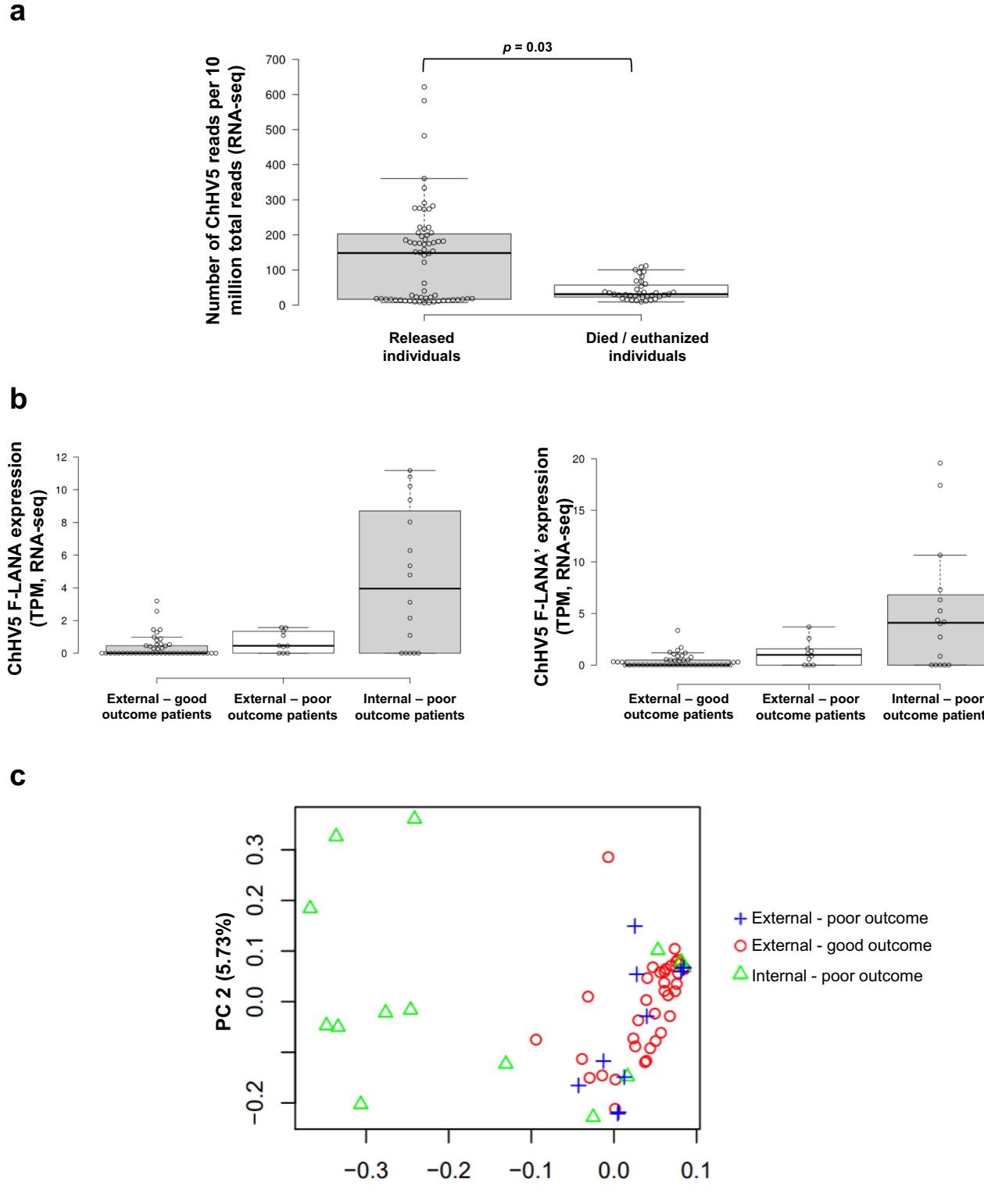

**Fig. 4 Patient rehabilitation outcome and ChHV5 gene expression. a** Box plot with Tukey whiskers of the number of ChHV5 reads (RNA-Seq) per 10 million total reads between patients based on outcome (released vs died in care/humanely euthanized). Individual sample values within each sample type are shown by the open points. Significant difference in averages between the two outcomes was determined by a Mann–Whitney *U* test and is denoted by an asterisk (*). *N* = 69 samples. Per outcome: released = 7 turtles; died/euthanized = 5 turtles. **b** Box plot with Tukey whiskers of the expression levels of the ChHV5 F-LANA genes (both copies of the F-LANA: F-LANA and F-LANA') in transcripts per million (TPM), by patient outcome and tumor location, as detected by RNA-seq. Individual sample values within each sample type are shown by the open points. **c** Principal component analysis (PCA) of the viral transcriptomes (minus 28 genes which had 0 reads across all samples, see methods) of all tumor samples (RNA-seq), by tumor location and patient outcome.

**Table 1 Viral genes differentially expressed between good versus poor outcome tumors (detected by EdgeR, quasi-likelihood pipeline analysis of RNA-seq).**

| ChHV5 gene | Gene length (bp) | $Log_2FC$ | p-value | FDR | External good outcome (average TPM) | External poor outcome (average TPM) | Internal poor outcome (average TPM) |
|---|---|---|---|---|---|---|---|
| F-ICP4a | 1955 | 1.542 | 4.06E−09 | 8.45E−08 | 0.05 | 0.08 | 1.14 |
| F-ICP4a' | 1955 | 1.498 | 3.81E−08 | 4.95E−07 | 0.07 | 0.22 | 1.08 |
| F-LANA | 1028 | 1.858 | 1.25E−09 | 4.35E−08 | 0.36 | 0.69 | 4.52 |
| F-LANA' | 1028 | 2.003 | 8.21E−11 | 8.54E−09 | 0.37 | 1.18 | 5.17 |
| F-lec1 | 626 | 0.924 | 5.99E−04 | 4.15E−03 | 0.06 | 0.17 | 1.46 |
| F-lec2 | 530 | 0.632 | 2.97E−02 | 1.47E−01 | 0.02 | 0.18 | 0.86 |
| F-M04 | 812 | 0.735 | 6.88E−03 | 3.58E−02 | 0.09 | 0.00 | 0.98 |
| F-Nec2 | 1739 | 0.830 | 3.12E−03 | 1.80E−02 | 0.00 | 0.03 | 0.43 |
| F-Sial | 962 | 0.730 | 6.85E−03 | 3.58E−02 | 0.08 | 0.06 | 0.96 |
| F-ULO.5 | 911 | 0.830 | 3.12E−03 | 1.80E−02 | 0.01 | 0.05 | 0.76 |
| HP3 | 326 | 1.400 | 1.46E−07 | 1.26E−06 | 0.71 | 1.46 | 7.64 |
| HP3' | 326 | 1.60 | 1.06E−07 | 1.22E−06 | 0.90 | 1.60 | 11.05 |
| HP30 | 536 | 0.836 | 2.41E−03 | 1.57E−02 | 0.04 | 0.00 | 1.63 |
| HP32 | 1685 | 1.500 | 3.43E−08 | 4.95E−07 | 0.03 | 0.07 | 1.30 |
| HP33 | 461 | 1.129 | 1.58E−05 | 1.27E−04 | 0.06 | 0.00 | 2.54 |
| HP34 | 473 | 1.287 | 1.37E−07 | 1.26E−06 | 0.12 | 0.84 | 3.25 |
| HP35 | 1439 | 1.793 | 2.15E−10 | 1.12E−08 | 0.02 | 0.23 | 1.48 |
| HP36 | 452 | 0.951 | 3.86E−04 | 2.87E−03 | 0.10 | 0.22 | 2.02 |
| HP37 | 2093 | 1.850 | 1.86E−09 | 4.83E−08 | 0.02 | 0.02 | 1.36 |
| HP4 | 662 | 1.674 | 9.75E−09 | 1.69E−07 | 0.15 | 0.24 | 4.32 |
| HP4' | 662 | 1.407 | 1.19E−07 | 1.24E−06 | 0.18 | 0.14 | 3.09 |

Note that differences in expression are primarily driven by changes in internal tumors. Gene names correspond to the ChHV5 reference genome [GenBank accession number: HQ878327.2][65].

(one FP and one non-tumor sample), while the highest viral DNA load observed was 3,673 RPTM. Interestingly, a high viral DNA load did not equate to high viral transcription (RNA) in the same sample (Table 2). As expected, WGS reads were dispersed across the entire ChHV5 genome (Supplementary Fig. 3a), not being restricted to ChHV5's transcriptionally active regions (Supplementary Fig. 1b). This confirms that the limited transcriptional signature is not due to a sequencing artifact. Together, the ChHV5 genome-level and gene-level TPM analysis highlight the marked difference in reads between viral DNA presence and viral RNA transcription in FP tumors. Conversely, the range of viral DNA and viral RNA within non-tumor tissue was largely overlapping suggesting that the ChHV5 present in non-tumor samples may be more likely to be transcriptionally active (Table 2 and Figs. 3a, 5a).

**Lung and urinary bladder tumors tend to have high viral loads**. We next examined the ChHV5 viral loads (viral DNA-based qPCR, ChHV5 *UL30* assay) in a wider cohort of internal tumor and matched non-tumor tissue types. Liver and kidney FP tumors had consistently lower viral loads than was seen in other tumor types, although wide ranges in ChHV5 loads were observed in other tumors (Fig. 5c), and some sequenced kidney tumors (all three from patient "Yucca") had high viral loads (Fig. 5a). Lung and urinary bladder FP tumors consistently had the highest viral loads (Fig. 5c), while, of the non-tumor tissue types assayed, bladder tissue also showed high viral loads, with detected levels being higher than in some of the FP tumors (Fig. 5c).

**ChHV5 may be vertically transmitted from mother to offspring**. Interestingly, the range of ChHV5 detected in hatchlings (one *C. mydas*, one loggerhead [*Caretta caretta*] and one leatherback [*Dermochelys coriacea*]) overlapped the range of ChHV5 in FP-afflicted non-tumor tissue, the FP kidney tumor, and FP-afflicted blood plasma samples (Fig. 5a). Of the three species, the leatherback sample had the highest number of ChHV5 reads, with many reads aligned to non-coding regions of the ChHV5 genome (Fig. 5a and Supplementary Fig. 4a). ChHV5 DNA was also detected at low levels in six samples from a number of tissues (eye lid, neck, cloaca, front flipper, and heart) from a deceased unhatched green sea turtle (Supplementary Fig. 4b), using the ChHV5 *UL30* qPCR assay (viral DNA range: 0–2.24E−05 pg/µl. Amplification ratio: 0.125). No ChHV5 DNA was detected from samples of brain, yolk, and intestine from this turtle.

**WGS reveals no evidence of papillomavirus (PV1) association with FP tumors**. We used our unbiased WGS to assess whether green turtle or loggerhead specific papillomaviruses (CmPV1 and CcPV1) were present in any of our samples, as although present in sea turtles CmPV1 and CcPV1 have not been strongly associated with FP tumors[66–68]. No CmPV1 reads were detected in any tumor samples, skin samples or plasma samples of FP-afflicted turtles (Supplementary Table 3). Furthermore, unlike ChHV5 (Fig. 3a), no shed CmPV1 or CcPV1 was detected in tank water of pooled FP-afflicted juvenile green and FP non-afflicted loggerhead hatchlings (Supplementary Table 3). From all 38 sequenced samples only two reads were detected for CcPV1, both in *C. mydas* kidney FP tumors from different individuals (Supplementary Table 3). This might be an artifact as the mate pair of these reads (paired-end sequencing), did not align to CcPV1. These two kidney tumors had 75 and 3,127 ChHV5 RPTM respectively (Table 2 and Supplementary Data 1), but only 0.037 and 0.014 CcPV1 RPTM respectively (Supplementary Table 3). Therefore, our WGS provides no evidence for a correlation between the presence of CmPV1 or CcPV1 and FP tumors, from green turtles from Floridian waters.

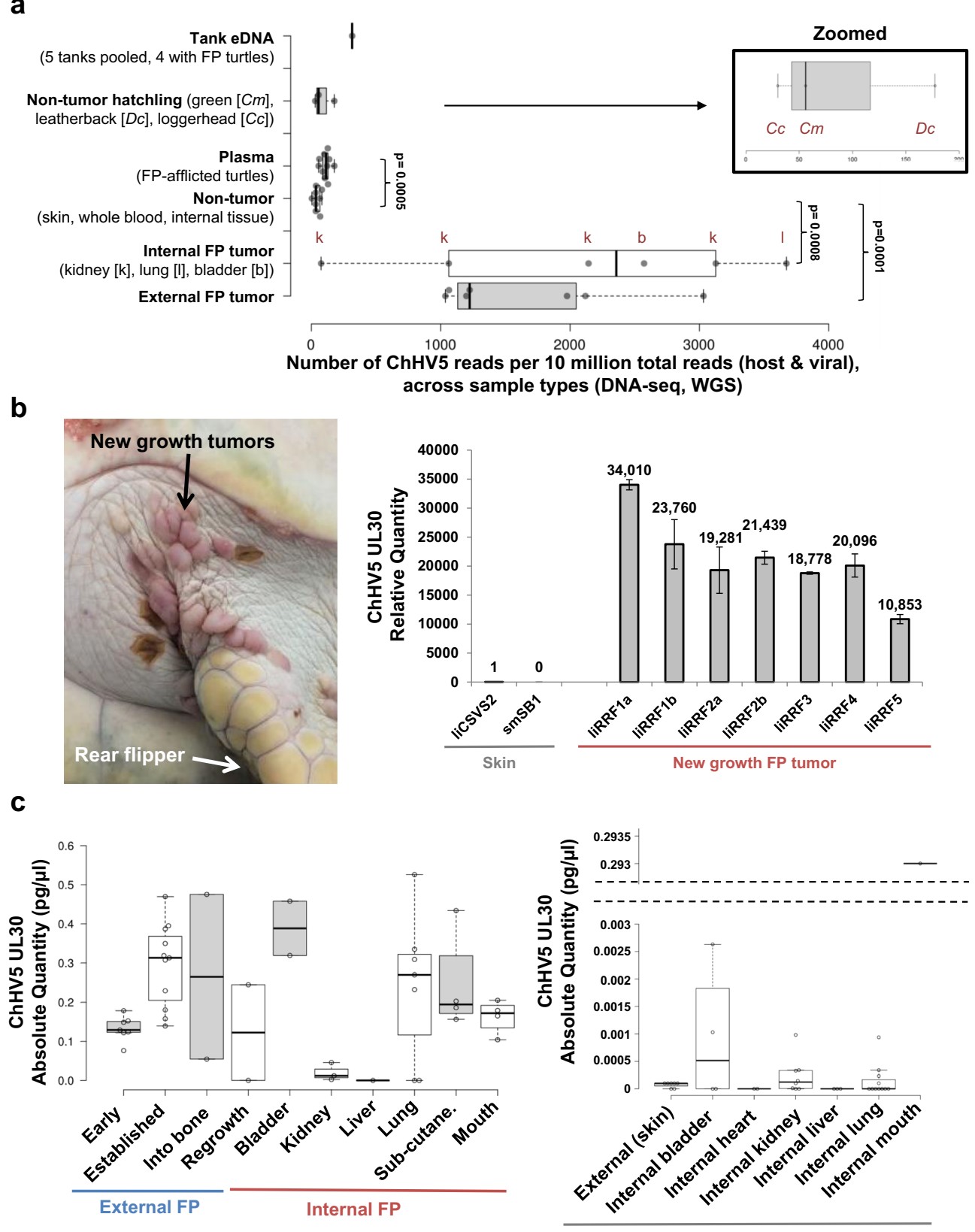

## Discussion

Anthropogenic activities are fueling the acceleration of the sixth mass extinction event[69], and human-wildlife conflict and intensive farming pressure are facilitating zoonotic disease pandemics (such as Ebola, SARS, and COVID-19) with the transmission of animal viruses to humans[29,70–74]. While the rapid environmental changes induced by human activities are increasing viral transmission from animals to humans, they are also altering the

**Fig. 5 Variation in ChHV5 DNA load across tumor and non-tumor tissue types, as assessed by whole-genome sequencing and qPCR. a** ChHV5 abundance (WGS DNA-based) across each sample type. Graph of the number of ChHV5 reads (DNA-seq) per 10 million total reads of 35 samples, with Tukey whiskers. Per sample type: external fibropapillomatosis tumor $n = 7$, internal fibropapillomatosis tumor $n = 6$ (4 kidney FP, 1 bladder FP and 1 lung FP), non-tumor $n = 8$ (2 skin, 2 kidney, 1 bladder, 1 lung, and 2 whole blood), plasma from FP-afflicted turtles $n = 10$, tissue from non-tumor hatchlings $n = 3$ (1 green, 1 loggerhead, and 1 leatherback), tank environmental DNA (eDNA) $n = 1$ (pooled sample of eDNA from 4 tanks with FP-afflicted juvenile green patients and 1 tank with FP-free loggerhead hatchlings). *P*-values for all groups with significant differences (*t*-test) to the non-tumor (skin and whole blood) sample cohort are shown on the graph. All samples are from green sea turtles, with the exception of two of the hatchling samples (loggerhead and leatherback), and the eDNA tank sample being pooled from four green and one loggerhead tank water extractions (Supplementary Fig. 2c). **b** Image of new growth tumors on patient "Lilac" (25-2018-Cm). "Lilac" was admitted to the hospital without tumors, but with a number of leech bites which later developed into FP tumors. ChHV5 relative quantification of "Lilac" new growth tumor samples and non-tumor skin punch biopsy samples using the *UL30* DNA qPCR assay[37] (Supplementary Table 6). Error bars denote the standard deviation of three qPCR technical replicates. ChHV5 DNA was detected in these growths via qPCR prior to them being classified as fibropapillomatosis by histology. **c** ChHV5 viral quantification box plot with Tukey whiskers of a range of FP types ($n = 43$ samples) and non-tumor tissue ($n = 36$) samples from 13 individual patients using the *UL30* DNA qPCR assay[37] (Supplementary Table 6). Individual sample values within each sample type are shown by the open points. Absolute quantity of ChHV5 was determined through a standard curve of known amounts (in picograms) of a *UL30* gene fragment (Supplementary Table 6). Truncated *x*-axis label, internal sub-cutane. = internal sub-cutaneous.

**Table 2 Comparison of viral sequencing reads per tumor and non-tumor tissue at the DNA and RNA level, for patients 27-2017-Cm and 25-2018-Cm "Lilac".**

| Sample | DNA viral load (RPTM) | RNA viral reads (RPTM) |
|---|---|---|
| *Patient 27-2017-Cm:* | | |
| Lung FP tumor | 3673 | 68 |
| Lung non-tumor | 68 | 36 |
| Kidney FP tumor | 75 | 30 |
| Kidney non-tumor | 66 | 26 |
| *Patient 25-2018-Cm ("Lilac"):* | | |
| External new growth FP tumor | 1036 | — |
| Skin non-tumor | 81 | — |

Number of reads per 10 million total reads (RPTM) are shown for each sample.

consequences of infections in animal populations, which are more susceptible to disease in the absence of healthy habitats and natural population sizes and ranges with their associated robust genetic variation[29]. Anthropogenic activities are likely increasing cancer rates in wild populations[75,76], with increased incidence and range of sea turtle fibropapillomatosis appearing to be a direct result of human activities impairing the ability of inshore juvenile sea turtles to combat the ChHV5 virus[3,31,77]. However, studies establishing functional links between environmental contaminants and fibropapillomatosis tumorigenesis, especially across large spatial scales, are required.

It is thought that environmental changes are key to conferring oncogenicity upon ChHV5, potentially through compromising the immune system of sea turtles[3,10,78,79]. While hosts attempt to mount an immune response[30] tumors still develop in a large proportion of individuals[3,5,9]. The causal link between ChHV5 and fibropapillomatosis also requires further scrutiny. As revealed here, either the role of ChHV5 in driving FP tumor formation and progression is restricted to extremely early tumorigenesis events, or it is driven by latently expressed genes, or is overstated.

As a number of ChHV5 strains exist, it was initially hypothesized that some strains were more oncogenic than others, thus accounting for differences of the severity of FP disease between individuals and populations. However, it has consistently been shown that different ChHV5 strains bear no correlation to disease severity[3,41,44,46,80], although large scale geographic differences do exist.

Similar to smaller cohorts of external FP tumors assessed by RNA-seq[5,81], we found no evidence to support widespread lytic ChHV5 viral replication within any of the tumor types profiled here. This includes early stage new growth tumors, suggesting that the viral 'hit and run' hypothesis[82] of oncogenic transformation does not apply to fibropapillomatosis tumors, or that if it does it must be extremely temporally restricted, likely before a visible tumor develops. Furthermore, our results reveal that if ChHV5 is responsible for inducing de novo tumor formation in internal organs, then the limited expression and load of ChHV5 in some internal tumors is highly irregular, particularly in kidney and liver tumors. That even tumor samples with high ChHV5 loads (DNA-level) have only low ChHV5 gene expression (RNA-level), suggests that rapid proliferation of host tumor cells infected with latent virus is the primary driver of the high viral loads sometimes observed in FP tumors. Fibropapillomatosis cell proliferation can be rapid and tumors can double in size in less than two weeks[1]. Similarly, the paucity of elevated levels of viral transcripts across all tumor samples, in addition to the lack of inclusion bodies, suggests that lytic virus may not be driving the establishment of numerous primary tumors within the same individual. The presence of inclusion bodies in FP tumors is relatively rare. In one study, from the analysis of 381 FP tumors from 17 individuals, epidermal intranuclear inclusion bodies were only identified in 35% of FP-afflicted individuals, and even within that subset of individuals, only 7% of their tumors contained at least one inclusion body[48]. Similarly, in situ hybridization of the ChHV5 UL30 gene only detected expressed ChHV5 transcripts in three out of 25 tumors[83]. Even within those three tumors, there was only a low level of expression which was spatially restricted[83]. Processes occurring in the dermis (especially relating to fibroblasts) may also influence the development of tumors and host-viral interactions, as FP tumors involve changes to both epidermal and dermal cells. Our whole-genome viral transcriptomics confirms that lytic ChHV5 gene expression is lacking in both external and internal tumors. Even while in their latent stage oncogenic viruses can manipulate host cell signaling and contribute to oncogenic transformation and tumor development[84–86].

Counterintuitively, we show that animals with lower overall viral transcriptional expression had worse rehabilitation outcomes than those with higher viral transcription. Oncogenic viruses can use latency as an immune avoidance strategy, with bouts of temporally and spatially restricted lytic activity helping to prevent a largescale immune response. This feature of oncogenic viruses may help to explain why ChHV5 is predominantly latent in FP tumors, and why lytic viral replication is so spatially restricted[48,83]. In this context, FP tumors with higher ChHV5 expression levels may be more prone to inducing immune responses, leading to improved patient outcomes. Further supporting this hypothesis, we recently showed that, in the same

patient cohort used in this study, higher expression levels of immune-related host genes were associated with better patient outcomes[38].

The latency-associated gene LANA and the four atypical ChHV5 genes (F-lec1, F-lec2, F-sial, and F-M04)[65] were more strongly expressed in internal tumors than external tumors, although overall expression levels were relatively low. The differing ChHV5 gene expression profiles between internal and external FP tumors, suggests that the mechanisms driving internal tumor development are different from those driving external tumor development. Supporting this, we recently showed that the host oncogenic signaling events driving internal and external tumor development also differ dramatically[38].

Sea turtle papillomaviruses (CmPV1 and CcPV1) were originally described in non-FP proliferative skin lesions[66,67]. However, a subsequent report described papillomavirus in cell lines derived from FP tumors and postulated that they may be associated with sea turtle fibropapillomatosis tumors[68]. Fibropapilloma tumors are associated with papillomaviruses in other domesticated and wild species[87–89], as well as a variety of human cancers, including cervical, and head and neck cancer[90,91]. However, it remains to be resolved whether papillomaviruses have any correlation to FP tumor occurrence. Using qPCR, CmPV1 was recently detected in skin tumor samples (<10 tumors) of Australian *C. mydas*, while cloacal swabs, blood, and normal skin samples from the same animals tested negative[68]. However, when we assessed for the presence of PV1 by WGS we detected no CmPV1 in tumor samples, skin samples, blood plasma or tank water. While a greater number of individuals and locations should be assessed, we found no evidence for PV1 as a causative or even correlative factor to sea turtle fibropapillomatosis, despite the prominent role of papilloma viruses in papilloma tumors of other species[87–91]. Given their small genome and ability to induce oncogenesis at lower loads than herpesviruses, targeted qPCR approaches should also be employed, to confirm whether papillomaviruses are absent from FP tumors[68,92]. Recently, such approaches detected PV1 in 47% of FP tumors in Australia[93].

ChHV5 DNA was identified in *C. mydas*, loggerhead (*Carretta carretta*), and leatherback (*Dermochelys coriacea*) hatchlings, and their abundance overlapped the range (no significant difference) seen in non-tumor tissue and blood plasma from FP-afflicted turtles, and an FP kidney tumor. Although it should be noted that ChHV5 genome coverage was not as complete in these hatchling samples as for tumor samples (Supplementary Fig. 4a). ChHV5 DNA was also detected at low levels, by qPCR, in a pre-hatchling that died prior to successfully hatching. A key objective should be to confirm whether vertical transmission of the virus occurs from mother to offspring. This hypothesis conflicts with the predominant hypothesis that ChHV5 infection is acquired only after juvenile turtles recruit to nearshore habitats[46,94]. If widespread vertical transmission is confirmed, the relative contribution of vertical versus horizontal transmission needs to be determined. If hatchlings are already infected with ChHV5 (vertical transmission), this has serious implications for any potential population-level vaccination-based mitigation strategies. On-beach (immediate nest emergence[95]) sampling and qPCR and sequencing-based ChHV5 detection should be conducted to confirm this finding and to determine the prevalence of ChHV5 infected hatchlings. Non-invasive sampling of sand from nest chambers may also reveal whether ChHV5 is present, given that we could detect ChHV5 from sand samples exposed to FP-afflicted patients.

In terms of horizontal vector-borne transmission, we show that ChHV5 was detectible in 90% of leeches removed from FP tumors, this contrasts to a reported 27.5% of leeches from FP-afflicted turtles testing positive for ChHV5[96]. This difference likely arises from relatively rapid testing (3 days to 7 months) in a rehabilitation setting, as opposed to the historical frozen sample set that was used for the Rittenburg et al. 2021 study[96]. Leeches likely only represent a mechanical vector of ChHV5 transmission, as studies to determine whether leeches function as a true vector with ChHV5 replicating within leeches (as an intermediary host) have not yet been conducted. While a broad range of ChHV5 quantity was detected across leech and patient tank water samples, ChHV5 tended to be more abundant in the water column than in marine leeches. This suggests that both direct water-borne transmission, and vector-borne transmission of ChHV5 are likely.

Accurate detection and monitoring of wildlife pathogens (both vertical and horizontal transmission) with the capacity to impact species survival is essential to devise and implement appropriate mitigation policies[63]. Environmental DNA approaches have been shown to detect aquatic pathogens earlier than traditional methods and provide advanced warning of infection and mass mortality events[57,63]. Our eDNA-based detection of ChHV5 in sea water is particularly significant given that, unlike more immediate acting pathogens, for virally-induced cancers there is generally a long lag time (years or decades) between infection and tumor formation[97–99]. It has previously been asserted that only a small percentage of FP tumors shed virus (7% of tumors in 35% of individuals)[48]. However, our eDNA-based monitoring of viral shedding demonstrated that ChHV5 could be detected even in the tanks of turtles with low tumor burdens and that ChHV5 shedding positively correlated to tumor burden. Other disease-associated green turtle herpesviruses have been shown to remain infective after exposure to sea water, though this has not yet been assessed for ChHV5[100]. Our findings regarding the predominance of latent ChHV5 within fibropapillomatosis would suggest that not all viral DNA in patient tank water may derive from lytically produced virus. Future studies should determine what proportion of detectible ChHV5 eDNA in patient tanks is derived from viral DNA within host cells compared with free viable infective virus. That urinary bladder samples had high viral loads has potential implications for the spread of ChHV5, as urine is another potential source of ChHV5 release from the body and transmission[49,50].

In the near future, eDNA technologies[57,63] may allow for the early detection of ChHV5 presence in the environment of vulnerable populations (akin to our detection of ChHV5 in the larger fishpond), and enable further research into the etiology, host species transmission and disease ecology of FP. This unique tool to dynamically track environmental viral shedding in response to disease progression and clinical interventions such as surgery and drug treatment will greatly enhance our ability to address fundamental questions relating to this disease, as well as to design evidence-based rational management strategies (e.g., containment/isolation policies). The approach can also be used for regular or intermittent monitoring of tank water in captive turtle populations, for early warning and disease prevention strategies. Such approaches can also improve our understanding of how environmental and clinical factors influence viral release and spread. The ability to perform eDNA-based quantitative viral shedding monitoring will also further enhance the utility of FP as a model not just to better understand this wildlife epizootic, but also to address unresolved questions relating to viral shedding for other animal and human pathogen-induced cancers.

**Summary**. Taken together, our results provide transcriptome and genome-level profiling of ChHV5 across external, new growth, established, post-surgical regrowth, and internal visceral tumors (Table 3). They reveal the paucity of lytic ChHV5 replication within tumors, even those harboring high viral loads. High viral

**Table 3 Summary of key study findings and their potential implications.**

| Key Points | Implications |
|---|---|
| 90% of leeches which feed on FP tumors harbor ChHV5 | Marine leeches remain a potential viral vector, with the potential to infect naive individuals, and must be considered in any mitigation strategies. |
| ChHV5 is detectable and quantifiable from sea water | Direct transmission (without the need for intermediary vectors) is, therefore, a likely possibility, with implications for management and mitigation strategies both in rehabilitation facilities and wild populations. |
| ChHV5 levels in the water column correlate to tumor burden | Tumors themselves are the predominant driver of viral load in patient tank water. As tumors are surgically removed the level of ChHV5 in tank water reduces. |
| ChHV5 is predominantly latent (whole transcriptomics) across new growth, established, regrowth and internal tumors and lower ChHV5 gene expression is associated with poor patient outcomes | Highly lytic ChHV5 is not likely driving crucial early stage tumor initiation events, suggesting that the 'hit and run' hypothesis of viral oncogenesis[5], does not apply to FP. |
| | Host genes are primarily responsible for driving tumor growth, suggesting anti-viral drugs are unlikely to be an effective FP treatment strategy. Atypical ChHV5 genes (F-lec1, F-lec2, F-Nec, F-Sial, and F-M04) are more highly expressed in internal tumors, though genome-wide ChHV5 expression is lower. |
| High viral loads in FP tumors are primarily due to latent, not lytic, viral replication | FP tumors with high ChHV5 loads (DNA-based detection), still have low viral expression levels (RNA-seq). |
| Latency of ChHV5 may be an immune evasion strategy | This immune avoidance strategy likely limits the potential of anti-viral drugs as FP treatments, and reduces the capacity of host immune systems to combat the infection. FP tumors with higher ChHV5 expression levels may be more prone to inducing immune responses, leading to improved patient outcomes. Further supporting this hypothesis, we recently showed that, in the same patient cohort used in this study, higher expression levels of immune-related host genes were associated with better patient outcomes[38]. |
| Hatchlings may already be exposed to ChHV5 | The detection of ChHV5 at levels similar to or higher than those seen in non-tumor tissue samples from FP-afflicted turtles suggests ChHV5 transmission may occur early in turtle lifecycles. The prevalence in hatchlings and whether transmission is vertical (from mother), occurs in nest, or upon emergence should be determined. |
| ChHV5 'super-spreaders/super-shedders' are unlikely, rather larger tumor volumes shed more virus into the water column | A higher tumor burden leads to more viral shedding, but this is driven primarily by larger tumors producing a similar level of virus per area as new growth tumors, rather than established tumors producing more lytic virus per area of tumor. There is no significant shift towards lytic virus production in established tumors, rather levels of lytic virus remain relatively low, but the sheer increase in tumor volumes results in more shedding. Therefore, any heavily burdened individual is likely to be shedding large quantities of virus. |

loads, where they do occur, likely represent latent virus being produced as infected tumor cells rapidly proliferate. They also revealed that a number of tumors, particularly some internal types, have extremely limited viral loads (DNA-based assessment), calling into question the role of ChHV5 in the establishment and growth of these tumors. The application of novel eDNA-based pathogen monitoring revealed that ChHV5 is shed into the water column, and quantification of ChHV5 eDNA revealed that the level of virus in patient tank water correlates to patient tumor burden. ChHV5 shedding is prevented upon excision of fibropapillomatosis tumors. Improved understanding of the oncogenic role of ChHV5 and its transmission can inform population management and clinical management strategies, including informing biosecurity and quarantine practices.

## Methods

**Tissue sampling.** Sampling was carried out under permit number MTP-21-236 from the Florida Fish and Wildlife Conservation Commission and with ethical approval from the University of Florida's Institutional Animal Care and Use Committee (IACUC). External fibropapillomatosis (FP) tumors were surgically removed by laser resection as part of the turtle's rehabilitative care or obtained during necropsy[5]. Non-tumored areas of the same turtles were obtained by 4 mm punch biopsies during the tumor removal surgery, or during necropsy. Non-tumored sites were selected by gross examination of the region by the attending veterinarian and confirmed visually to be tumor-free normal skin regions and not bordering any tumorous regions by the attending vet, technicians, and researchers.

External tumors were sampled from a range of body locations, including skin, plastron, and eye tumors (Supplementary Data 1). Internal tissue samples (tumors and non-tumor tissue samples from lung, kidney, bladder, liver, mouth, sub-cutaneous, and heart) were obtained from animals during necropsies conducted immediately after euthanasia. Note that no animal was euthanized for the purposes of this study, but rehabilitating sea turtles found to harbor internal tumors are currently euthanized in Florida, according to Florida Fish and Wildlife Conservation Commission protocols, as no treatment yet exists for internal tumors. Additional complications arising from surgery and other health concerns may also sometimes necessitate the humane euthanization of sea turtles in rehabilitation. Internal tissue samples were treated the same as the external samples. All samples were obtained from juvenile *C. mydas*, as this life stage is the most commonly afflicted by the disease. Sex is not readily determinable in juveniles, but was provided for individuals that were euthanized due to internal tumors or other complications and in which necropsies were performed, or for individuals that were endoscoped (KARL STORZ, Multi-Purpose Rigid Endoscope for small animals) as part of their rehabilitative care (see Supplementary Data 1). Samples were stored in RNA-later (Qiagen) at −80 °C, according to the manufacturer's instructions, until extraction. Samples were stored between <1 day and 8 months, with 85% of samples being extracted within one month.

For the three hatchling samples (one green, one leatherback, and one loggerhead) utilized for WGS, these were admitted to our rehabilitation hospital but died in care. They were housed completely separately to FP-afflicted juvenile patients and frozen at −20 °C upon death. Within two months the carcasses were thawed and during necropsy a cross section of the front flipper of each hatchling was used for DNA extraction (see below). This sampling was conducted using sterile technique, with separate instruments (single use disposable), and was conducted in a lab which never housed FP-patients or any patients from other size classes. For the unhatched green sea turtle utilized for qPCR, upon death this animal was necropsied (on the same day) and tissue samples were obtained from

the following locations; eye lid, neck, cloaca, front flipper, heart, brain, yolk, and intestine, and DNA extracted immediately (see below). Prior to death, this turtle remained in its shell and never encountered FP-afflicted turtles, areas, or instruments.

Leeches were removed from six patients during routine admission exams and stored in RNAlater (Qiagen) at −80 °C until extraction. Leeches used for pooled leech extractions were stored for up to 7 months prior to extraction (from "Ruth Gates", 07-2015-Cm and 09-2015-Cm were stored for 1 month, from "Bruno Hofer" for 5 months, and from "Richard Dawkins" for 7 month). Leeches used for individual leech extractions (from "Broccoli") were stored for 3 days prior to extraction. Leeches were removed from FP tumors, from non-tumor sites (soft skin/plastron) of FP-afflicted turtles, or from non-FP afflicted turtles. Each entire leech (individual samples) or groups of approximately 10 entire leeches (pooled samples, from the same location type) including the blood pellet were homogenized and used for DNA-extraction (using a DNeasy Blood and Tissue Kit [Qiagen, Cat No. 69504], see below). The majority of the leeches were removed from inguinal regions (both tumor and non-tumor), all 30 leeches extracted individually were removed from two FP tumors on patient "Broccoli's" ventral inguinal region (Fig. 1a).

### RNA and DNA Isolation, library preparation, and sequencing from tissue and eDNA samples.
All sequencing was conducted at the University of Florida's Interdisciplinary Center for Biotechnology Research Core Facilities. For RNA-Seq samples, total RNA was extracted using either an RNeasy Fibrous Tissue kit (Qiagen, Cat No. 74704) or RNeasy Plus kit (Qiagen, Cat No. 74134) with column-based genomic DNA removal, according to the manufacturer's instructions. Ninety RNA samples, comprising 70 FP tumor samples and 20 non-tumor samples from 12 juvenile green turtles which had stranded in Northern Florida, were used for sequencing. Samples were further categorized by tissue type, as well as growth profile for the external tumors only (see Supplementary Data 1). Sequencing libraries were generated from 500 ng of total RNA using the NEBNext Ultra RNA Library Prep Kit for Illumina (New England Biolabs, Cat No. E7530), including polyA selection, according to manufacturer's protocol. Size and purity of the libraries were analyzed on a Bioanalyzer High Sensitivity DNA chip (Agilent). The RNA samples used for library construction had a RIN value range of 7.2 to 9.8, with the median RIN value of all samples being 9.1. Libraries were sequenced as paired-end reads with a read length of 100 bp on a HiSeq 3000 (Illumina). ERCC Spike-In Mix (ThermoFisher) was used as an internal control: 2 μL of 1:400 diluted ERCC Spike-In Mix with 500 ng of total RNA input.

Fibropapillomatosis-afflicted juvenile *C. mydas* tissue DNA-seq samples were sequenced in two batches, 6 initial samples on a HiSeq 3000, and 18 NovaSeq 6000 (Illumina) samples. For HiSeq 3000 samples, DNA was extracted using a DNeasy Blood & Tissue Kit (Qiagen, Cat No. 69504). Libraries were generated using Illumina TruSeq DNA PCR-Free Library Prep kit including fragmentation with a Covaris S220 sonic disruptor. Size and purity of the libraries were analyzed on a Bioanalyzer High Sensitivity DNA chip (Agilent). Libraries were sequenced as paired-end reads with a read length of 100 bp on an Illumina HiSeq 3000. Six whole genomic DNA samples, comprising three FP tumor samples and patient-matched non-tumor samples from two juvenile green turtles which had also stranded in Northern Florida, were used for sequencing. These samples were further categorized by tissue type, with one tumor and patient-matched healthy tissue sample each coming from an external, kidney, and lung tissue source (see Supplementary Data 1). For NovaSeq 6000 samples (Supplementary Data 1), the 18 DNA samples were fragmented on a Covaris S220 sonicator, to generate ~300 bp fragments. Fragmentation was followed by AMPure clean-up (0.9:1.0 beads:sample ratio). The cleaned-fragmented material (50 ng) was used as input for library construction. Sequencing libraries were constructed using the NEBNext UltraTMII DNA Library Prep Kit for Illumina (Cat# E7645S), according to the manufacturer's protocol. A set amount of 10–20 ng of sample was used to barcode and generate full adapter-ligated libraries through PCR. PCR was done for only 6–7 cycles of amplification in order to minimize duplicate reads. Barcoding was done using the indexing reagents provided in the NEBNext Unique Dual Index Oligos kit (Cat# E6440S). The final libraries were quantified by fluorescence (QUBIT, ThermoFisher), and sized on the Agilent TapeStation (DNA5000 Screen Tape). A yield of 50–60 ng of library was obtained of an average size ranging from 460 to 600 bp. Libraries were normalized and pooled equimolarly. This step was followed by treatment with the "Free Adapter Blocking Reagent" protocol (FAB, Cat# 20024145) in order to minimize the presence of adaptor-dimers and index hopping rates. The library pool was diluted to 2.5 nM and sequenced (one S4 lane, 2 × 150 cycles [paired-end]) according to Illumina NovaSeq6000 sequencing protocol, using 180 pM loading concentration and 1% PhiX spike-in control. Approximately 10 billion paired-end reads were obtained for the entire run (Q30% > = 80%; Cluster PF = 70%), ~8.7 billion paired-end reads were generated (NovaSeq 6000) for the 18 *C. mydas* samples, with the remainder of the reads relating to samples from other projects.

In addition, whole genomic DNA was extracted from 10 plasma samples taken from six individual turtles during the course of their rehabilitation (Supplementary Data 1). Excess blood was utilized from blood samples drawn for routine medical care by our veterinarians and plasma was separated from the red blood cells by centrifugation. Only 100 μl of plasma was collected and stored as allowed by permit

number MTP-21-236 from the Florida Fish and Wildlife Conservation Commission. DNA was extracted from 60 μl of plasma using a DNeasy Blood & Tissue Kit (Qiagen, Cat No. 69504) and used to produce low input Illumina Fragment libraries using a Low Input Library Prep kit v2 (Clontech Laboratories, Inc., Catalog No. 634899). DNA was fragmented using the Covaris S220 sonic disruptor and libraries were then sequenced as paired-end reads with a read length of 100 bp on an Illumina HiSeq 3000.

Furthermore, three whole genomic DNA samples were also taken from ground flipper tissue samples from three deceased hatchling turtles (never in contact with FP-afflicted patients or water/tanks) of each separate species: a green (*C. mydas*), loggerhead (*C. caretta*), and leatherback turtle (*D. coriacea*), and processed following the methods for tissue (HiSeq 3000) as detailed above. The leatherback sequence length was the only sample that differed, with paired-end reads of length 150 bp instead of 100 bp. TPM generation of hatchling samples was done using viral aligning reads only, as not all of the turtle species have published reference genomes.

Finally, one pooled library of environmental DNA (eDNA), combining five holding tank water samples from the Whitney Laboratory Sea Turtle Hospital facility of the University of Florida was also used for sequencing. For this pooled sample, water collection, filtration, and extraction were performed separately on each tank sample, and the final purified DNA was pooled prior to library preparation. Pooling was conducted to obtain the average ChHV5 load across the five tanks. Seawater from five tanks (four housing juvenile green sea turtles and one housing loggerhead post-hatchling washbacks, 500 ml seawater per tank) was filtered (EMD Millipore PES 0.22 μm Sterivex filter) by a hand pump and DNA was extracted from the filter using a Qiagen DNeasy Blood & Tissue Kit (Qiagen, Cat No. 69504) and modified extraction protocol was carried out[58], with an overnight (24–36 h) incubation of the filter at 56 °C in Buffer ATL and Proteinase K in a hybridization oven with agitation. Libraries were generated using a NEBNext Ultra II DNA Library Prep Kit for Illumina (New England Biolabs, Cat No. E7645), including fragmentation with a Covaris S220 sonic disruptor. Fragment size and purity of the libraries were analyzed on a Bioanalyzer High Sensitivity DNA chip (Agilent). Libraries were sequenced as paired-end reads with a read length of 100 bp on an Illumina HiSeq 3000.

### Quality control and read trimming.
The software FastQC—https://www.bioinformatics.babraham.ac.uk/projects/fastqc/—was used to assess data quality. Reads were then trimmed with trim_galore (The Babraham Institute, version 0.5.0) to remove ends with a Phred quality score less than 30, to remove adaptor sequences, and to remove sequences fewer than 25 bp after trimming. For any samples that contained overrepresented sequences according to FastQC, the trimmomatic tool[101] (version 0.36) was then used to remove these sequences from reads and any sequences less than 25 bp after trimming. The number of raw reads per sample and reads remaining after trimming can be found in Supplementary Data 1.

### Read alignment and read counts.
Reads from all samples (RNA-seq, DNA-seq, and eDNA) were first aligned to the ChHV5 genome [GenBank accession number: HQ878327.2][65] to determine the level of ChHV5RNA and DNA present in each sample using bowtie2[102] (version 2.3.5.1). The overall alignment rate to the ChHV5 genome was low for both RNA-seq and DNA-seq samples, with most reads aligning to the green turtle genome (NCBI GenBank Accession numbers: GCA_000344595.1 and GCA_015237465.1), as expected (Supplementary Data 1).

Transcript abundance for ChHV5 virus-specific transcripts was generated using htseq-count[103] (version 0.6.1p1) with the following parameters: not strand-specific, feature type 'gene', intersection non-empty mode, and a minimum aQual of 0. Count tables for viral transcripts were merged for all RNA-seq samples and counts were normalized for gene length and sequencing depth by transcripts per million (TPM) (Supplementary Data 2). TPMs were generated using the combined host (*C. mydas*[30]) and ChHV5 reads to allow direct comparisons between host and viral gene expression levels.

### Differential expression analysis.
Prior to differential expression analysis, a 1 was added to all gene counts to avoid zero counts in the analysis. The following viral genes that had the same counts across all samples were removed due to having a variance of 0 across samples: UL15A, UL15B, UL16, UL20, UL26.5, UL31, UL37, UL53, HP1/HP1', HP2/HP2', HP7, HP8, HP11, HP12, HP14, HP15, HP18, HP21, HP22, HP23, HP25, HP26, HP27, HP29, HP31, HP38. An external FP tumor sample, emLFF3H, was also removed from the analysis due to it being an outlier. A principal component analysis (PCA) plot (see Fig. 4c) was generated using the PtR script in the Trinity toolkit[104]. Differential expression analysis was carried out in the R statistical environment (v 3.4.4) using the quasi-likelihood algorithm within the software package 'edgeR' (v 3.20.9)[105–107]. Samples were first grouped by rehabilitation outcome: good outcome (released, $n = 46$ samples) versus poor outcome (died in care or humanely euthanized, $n = 23$ samples). The trimmed mean of M-values normalization method was then used to calculate effective library size. To model read counts for each gene, the quasi-likelihood extension for the negative binomial distribution[107] was used. Differential expression analysis was then conducted in which good outcome was set as the "treatment" and poor

outcome as the "reference" or "control" group. A false discovery rate (FDR) of 5% was used as a cut-off to determine significantly differentially expressed viral genes between rehabilitation outcomes. Results of this analysis are included in Table 1. Boxplots were generated using BoxPlotR[108].

**Tissue and eDNA qPCR.** qPCR assays were conducted on non-tumor tissue and FP tissue samples to quantify viral load within a range of tumor types, as well as water and sand eDNA samples to look at viral shedding dynamics within rehabilitation tanks. DNA was extracted from 79 tissue samples from 14 juvenile green turtles which had stranded in Northern Florida (Supplementary Data 3), 25 tissue samples from an unhatched green sea turtle and from leeches removed from six turtles as detailed above. Leeches from the same location were either pooled prior to extraction (~10 leeches per pool, Fig. 1d), or extracted individually (Fig. 1e, Supplementary Table 1). DNA was also extracted from 50 tank water samples (19 samplings with two-three sampling replicates per sampling event) and an additional 26 tank water samples from one prolonged-duration individual (13 samplings with 2 sampling replicates per sampling event). DNA was also extracted from 1 pond water sample (three sampling replicates), on the surface of the east end of the pond, close to the outflow pipe (Supplementary Fig. 1a). The fishpond is ~661,000 liters. Environmental DNA was extracted as per the eDNA sequencing samples above, with 500 ml seawater filtered per tank, except for the samples in Fig. 4e in which 1-liter seawater per sample was collected. Filtering (EMD Millipore PES 0.22 μm Sterivex filter) was performed by hand pump, except for samples depicted in Fig. 4e for which a Geotech Peristaltic Pump was used.

For sand eDNA samples, a 50 ml Falcon conical centrifuge tube was filled with sand from the base of a plastic holding box used to house an FP-afflicted juvenile *Chelonia mydas* ("Archie Carr" 49-2020-Cm) (on two separate occasions, approximately one month apart). The patient was housed in the box for 30 min, (dry-docked) during routine rehabilitative treatment at the Whitney Sea Turtle Hospital. While awaiting examination or receiving treatment, patients are routinely dry-docked for short periods of time. The 50 ml sand samples were sub-divided into 2–3 10 ml biological replicates. Each replicate was then washed in 20 ml 1× IDTE pH 8.0 TE Solution in individual 50 ml Falcon conical centrifuge tubes. Samples were shaken gently by hand then set on a rocking platform for 1 h at room temperature, shaking gently by hand every 15 min. Samples were then rested until sand sunk to the bottom of each tube (~30 s), then the aqueous layer was immediately pipetted into a 60 ml sterile BD luer lock syringe. Samples were hand filtered through 0.22 μm Sterivex-GP Pressure Filter Units and capped with B. Braun luer lock caps. 740 μl Buffer ATL and 60 μl Proteinase K from a Qiagen DNeasy Blood and Tissue Kit were added to each sample and eDNA was extracted as described for water eDNA samples (see above).

ChHV5 viral load was quantified using TaqMan Fast Advanced Mastermix (ThermoFisher, Cat No. 4444557) according to the manufacturer's protocol by amplifying the ChHV5 virus-specific DNA polymerase (*UL30*) gene[37] (Supplementary Table 4). A species-specific assay was also developed to target the 16S ribosomal RNA gene (rRNA) for Atlantic populations of *C. mydas* to serve as positive controls and to compare the level of viral DNA present against the level of *C. mydas* DNA present within each sample (Supplementary Table 4). This sea turtle species was the primary focus as they are the ones that are most commonly housed at the Whitney Sea Turtle Hospital and most commonly afflicted by FP. A LightCycler480 Instrument II (Roche) was used for amplification and cycling parameters were as follows: 95 °C 10 min, 45 cycles of 95 °C 10 s, 60 °C 20 s, and 72 °C 20 s, or an Applied Biosystems QuantStudio 6 Pro was used for amplification and cycling parameters were as follows: 50 °C for 2 min, 95 °C for 10 min, 45 cycles of 95 °C for 15 s and 60 °C for 1 min. All samples were run in triplicate with negative controls, except sand eDNA samples, fishpond water eDNA samples, leech samples, and unhatched tissue samples which were run with six technical replicates. Both absolute quantification and relative quantification methods were utilized in this study. Standard curves using synthetic fragments of the *UL30* gene and *C. mydas 16S rRNA* gene (see Supplementary Table 4) were generated to calculate the amount of DNA of these two genes present within each sample (in pg of DNA). For relative quantification, the ΔCt method was used, in which a single sample was set as a reference point to which all other samples were compared.

**Tumor burden calculations, statistics, and reproducibility.** Tumor burdens were assessed by total tumor surface area using ImageJ (https://imagej.nih.gov/ij/)[1]. ImageJ image analysis was used to make accurate measurements of tumor two-dimensional surface area, from images taken using an Olympus Tough TG-4 a~30 cm from each lesion, with a 25 cm scale bar for accurate pixel comparison.

For comparison of viral loads (Fig. 5a), data that was normally distributed was compared using *t*-tests, for comparison of data that were not normally distributed (even after log transformation), non-parametric tests (Mann–Whitney *U* Test) were performed. Significance was considered as $p \leq 0.05$. For correlation of tumor burden with viral shedding, and viral and green turtle eDNA a Pearson Correlation Coefficient test was used (Fig. 2a,f). The underlying tumor surface area and ChHV5 concentration data used for the correlation reported in Fig. 2f, is present in Supplementary Data 4. For comparison of viral loads among different groups of RNA-seq (Fig. 3a) samples (non-tumor, established FP, regrowth FP, new growth FP, kidney FP, and lung FP), a Kruskal–Wallis test followed by a

Dunn–Bonferroni post hoc to indicate significant differences among groups was performed in SPSS (IBM SPSS®).

**Histology methodology, embedding, sectioning, and staining.** Turtle tissue samples were surgically removed using a $CO_2$ laser and stored in 4% paraformaldehyde at 4 °C overnight. Samples were washed twice in 1× PBS for 10 min, once in Milli-Q $H_2O$ for 10 min, twice in 50% ethanol for 15 min, twice in 90% ethanol for 15 min, and twice in 100% ethanol for 15 min. Samples were then stored in 100% ethanol at 4 °C for 5 nights. Samples were washed in 100% aniline for 1 h, 50:50 aniline:methyl salicylate for 1 h, and twice in 100% methyl salicylate for 1.5 h. Samples were then stored in 50:50 methyl salicylate:paraffin at 60 °C overnight. Samples were washed twice in 100% paraffin at 60 °C for 3 h. Samples were then stored in 100% paraffin overnight. Finally, samples were embedded in 100% paraffin and stored at 4 °C. Paraffin blocks were sectioned into 6 μm ribbons of six tumor sections each, on charged Fisherbrand Superfrost Plus microscope slides using an AO Spencer '820' microtome and stored at room temperature.

For hematoxylin and eosin (H&E) staining, sectioned slides underwent a series of washes: two 7-min washes in xylene, two 7-min washes in 100% ethanol, two 7-min washes in 95% ethanol, 2 min in distilled $H_2O$, 2 min in hematoxylin, two 15-s washes in distilled $H_2O$, 2 min in distilled $H_2O$, 3 min in $NH_3$, 2 min in distilled $H_2O$, 2 min in 95% ethanol, 1 min in eosin, two 2-min washes in 95% ethanol, two 2-min washes in 100% ethanol, and finally two 2-min washes in xylene. Fisher Scientific Permount mounting media was added to stained slides, a cover slip was placed on top, and slides were left to dry overnight before imaging on the Zeiss M2 Axio Imager.

Features of tumors were evaluated by a veterinary pathologist for comparison with the molecular analysis (Supplementary Table 2). Sections were examined for intranuclear viral inclusion bodies. The cellularity of the dermal component of cutaneous tumors and internal tumors was subjectively scored, by the pathologist, on a scale of 1 (sparsely cellular) to 3 (densely cellular). The presence of lymphocytic inflammation within the tumors also scored from 0 (absent) to 3 (abundant). The presence of ulceration of the epidermis and associated heterophilic (granulocytic) inflammation also was noted. These features were selected as they may represent aspects of tumor development, regression, or host response relevant to our analysis.

**Reporting summary.** Further information on research design is available in the Nature Research Reporting Summary linked to this article.

## Data availability

The RNA-Seq and DNA-Seq data including raw reads are deposited in NCBI (https://www.ncbi.nlm.nih.gov/) under BioProject ID: PRJNA449022.

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

## Acknowledgements

Funding was generously provided by The Sea Turtle Conservancy, Florida Sea Turtle Grants Program under project number 17-033R, the Save Our Seas Foundation under project number SOSF 356, the National Save The Sea Turtle Foundation, Inc. under project name Fibropapillomatosis Training and Research Initiative, and a Welsh Government Sêr Cymru II and the European Union's Horizon 2020 research and innovation programme under the Marie Skłodowska-Curie grant agreement No. 663830-BU115. This research was also supported by Gumbo Limbo Nature Center, Inc d/b/a Friends of Gumbo Limbo (a 501c3 non-profit organization) through a generous donation through their Graduate Research Grant program and by an Irish Research Council Government of Ireland Postgraduate Scholarship, under project number GOIPG/2020/1056. Warmest thanks to Mark Q. Martindale, Nancy Condron, and the veterinary and rehabilitation staff and volunteers of the Sea Turtle Hospital at Whitney Laboratories. Thanks also are due to David Moraga, Yanping Zhang, and Mei Zhang of UF's Interdisciplinary Center for Biotechnology Research Core, Whitney Daniel and the staff of the South Carolina Aquarium, Elizabeth Ryan, Brian A. Stacy, and Nicole Stacy for informative discussions, and Florida Fish and Wildlife Conservation Commission's Meghan Koperski for valuable assistance with permitting.

## Author contributions

D.J.D. designed and supervised the project. J.A.F., D.J.D., K.Y., and L.W. generated the data. K.Y., J.A.F, D.J.D., C.S., P.L. D.R.R., and R.T. performed data and bioinformatics analysis. B.B., D.R.R., C.B.E., and R.T. provided veterinary care, including tumor removal surgeries. D.J.D., J.A.F., K.Y., and L.W. wrote the manuscript. J.W., C.S., and S.C. performed critical reading of the manuscript. All authors read and approved the final manuscript.

## Competing interests

The authors declare no competing interests. Additionally, the funding agencies had no role in the conceptualization, design, data collection, analysis, decision to publish, or preparation of the manuscript.
