## [Peer Review File · Communications Biology]

Reviewers' comments:

Reviewer #1 (Remarks to the Author):

This manuscript clearly reflects the final entering into the fibropapillomatosis genomics and eDNA-based pathogen monitoring. This field has been barely scratched until now, but it can potentially reveal the precise mechanisms through which the virus is transmitted and the role of ChHV5 in host cell transformation and tumor progression. I personally applaud the effort and work done in this study. Such research will provide insights into this wildlife epizootic and reveal how ChHV5 can rapidly induce novel cancer incidence on an epidemic scale. Such information is vital to enable improved management, treatment and mitigation strategies to be developed to combat this sea turtle conservation-relevant disease epizootic.

There are, however, a couple of technical issues and clarifications that I requested to be addressed. Due to the length, complexity and size of the dataset generated in this study, it is completely understandable that more than one publication will be achieved. However, it is also important to check how much overlapping of data and information in general is in between this complementary publication (in press or in review) and the present study. Besides these two big aspects, there are some minor comments, more sort of personal curiosity commentary to the authors (which I would love to chat about it for long periods of time) and suggested changes

I think if few of the issues that I highlighted are addressed and adjusted, I would definitely give my full recommendation to the Editors to accept this paper, which once more, I enjoyed a lot reading and I will look much forward reading in a final version, once fully accepted.

Reviewer #2 (Remarks to the Author):

This manuscript is chock-full of interesting data. The presentation of the work, however, needs additional editing. For example, there are several parts of the methodology that were either totally left out or abbreviated to the point where it was difficult to understand exactly what was done. This needs to be addressed. There were also some errors in allocation of descriptions to the wrong section- i.e., passages suitable for the discussion section were included in the results, etc. Also, the paper is so dense and multi-faceted to the point of detracting from the most important part, namely the isolation of ChHV5 in tank water samples and the important implications this could have. I recommend restructuring the presentation, adding in more details about the sampling methodology in particular, and highlighting the most relevant and novel parts of the study instead of burying them amongst data that merely supports other, previous studies. Detailed comments are below. Since there were no line numbers, comments are annotated based on sections and page numbers.

Title: Is there supposed to be a period at the end of the title?

Introduction

p.2:

- northeastern Florida, correct?
- here and throughout, replace "C. mydas" with "green turtles"
- "Long-lived reptiles have normally robust anti-cancer defenses^{24,25}, with the exception of FP, reports of neoplasia in sea turtles are rare." This sentence needs references for reports of non-FP neoplasia in sea turtles.
- change "exposures are likely impairing" to: "exposures likely impair..."
- remove comma after "While some environmental co-factors"

p.3

- "alphaherpesvirus" is just one word
- "released into the environment and" - Insert comma after "environment"

Results

p.4

- "To

investigate the hypothesis that ChHV5 might be latent in established tumors, but more active during crucial early stage tumor initiation events, and determine whether ChHV5 was lytic or latent in different tumor types, we sequenced 90 samples (RNA-seq) from five FP tumor types: new growth external (newly formed and unassociated with surgical sites), established external, regrowth external (post-surgical excision), internal lung and internal kidney. The resulting reads were then aligned to the ChHV5 genome⁵⁷." - These details belong in the Methods section.

p.5

- Here and throughout, the statistical methods (Kruskal Wallis, Mann-Whitney, correlation, etc.) described in the Results section were not even mentioned in the Methods section. Please reconcile.
- Does 'external FP' differentiate between tumors of the soft skin, eyes, and/or shell? Details on the origins of the tumors used in this study would be really helpful in understanding and interpreting these results.

p.7

- "This suggests that rapid proliferation of host tumor cells infected with latent virus is the primary driver of the high viral loads sometimes observed in FP tumors. Fibropapillomatosis tumors can double in size in less than two weeks¹." - This belongs in the Discussion section.
- at least at first mention, identify the "bladder" as the "urinary bladder"

p.8

- "These results have implications for the spread of ChHV5, asurine is a potential source of ChHV5 transmission^{43,44}." - This belongs in the Discussion section.
- Depending on the circumstances under which the hatchlings were sampled, this data may be problematic. If the hatchlings were housed in tanks at the rehab facility prior to dying, it is equally (if not more) likely that they were exposed through tank water than being exposed via vertical transmission. If they were housed in tanks prior to death, this should be addressed in the discussion section. If so, I suggest this section be minimized/discussed differently.
- "If hatchlings are already infected with ChHV5 (vertical transmission), this has serious implications for any potential population-level vaccination-based mitigation strategies. On-beach (immediate nest emergence⁵⁸) sampling and sequencing-based ChHV5 detection should be conducted to confirm this finding and to determine the prevalence of ChHV5 infected hatchlings." - This belongs in the Discussion section.
- The 2nd paragraph prior to the sentence starting with "Therefore," belongs in the Discussion section.

p.9

- This is by far the strongest finding of this study. I recommend that the manuscript be structured to lead with this result.
- This is the first mention that the leeches were sourced from FP tumors. This methodology should be described in detail in the methods section.
- What statistical test was used to analyze this correlation?

p.11

- As stated in my above comment, if the hatchlings were housed in tank water prior to death, these results are likely overstated and should be toned down.

p.12

- What about regular or intermittent monitoring of tank water in captive turtle populations, as a disease prevention strategy? I think this warrants mentioning here.

p.13

- Which non-tumor areas, specifically? Soft skin? From which body part(s)? Be more specific.
- What type(s) of non-tumor tissues were sampled? I.e., what organ system(s)? Be more specific.
- You may want to point out that the euthanasia of turtles with internal tumors is based on a protocol recommended by FWC.
- What about the hatchling turtle samples? Sampling of hatchlings needs to be described in detail here.
- Up to how many weeks/months were samples stored prior to DNA extraction?

p.15

- How were these hatchlings encountered/housed? Were they kept in tanks prior to dying? Or were they found dead in the nest or on the beach? If kept in tanks, were they exposed to tank water shared by juvenile green turtles in rehab?
- As written, this sounds like only a single tank sample was taken. The tank water sampling protocol needs to be described in more detail here.
- Why were these samples combined from different tanks? This needs to be explained and justified. Or maybe I am just confused by the way this is written. Either way, please clarify.

p.17

- What types of tissue samples? This is unclear. Sampling details on what tissue types, from what turtles, are needed.
- There is actually no description of the sampling of the leeches. Where were they removed from? From turtles with FP? Where on the turtles' bodies were they attached? Were the leeches ground up whole and DNA extracted from that? Or were they dissected and the guts subjected to extraction?
- As noted above, details should be clearly explained in the tank water sampling methods section, above.

p.18

- Was total tumor surface area compared to turtle size for a % of body affected? A brief description of this methodology is warranted.
- Were these slides reviewed by a veterinary pathologist? There should be a complete description of how the slides were reviewed.

Table 1

- Do these genes have associated GenBank accession numbers? If so, it would be good to include those in this table for reference.

Table 3

- Not sure this table is necessary, or maybe just include it as a supplement. It should suffice to summarize in the text that 0 reads were detected for all but one sample.

Figure 1B. This is interesting, in that it appears to somewhat contradict what you found with tank water, in that patients with large well-established tumors shed more virus into tank water than those with small new-growth tumors. If real, I think this contradiction should be addressed in the discussion.

Figure 2B. These data seem to suggest that the mechanisms driving internal tumor development are different from those driving external tumor development. This is touched on in the discussion but could be addressed more robustly, perhaps with examples/comparisons to better-known herpesviruses such as Kaposi's sarcoma.

Figure 3A. Why do the authors think this lung sample had such a high number of reads? Any clues about why this would be, based on known herpesvirus biological mechanisms? Please address in the discussion.

Figure 4A. This is mostly skimmed over in the manuscript, probably because it was not well described in the methods section. Please reconcile and describe in more detail if you are going to include it at all.

Reviewer #3 (Remarks to the Author):

General comments:

This paper provides useful genomic information and environmental information about fibropapillomatosis in sea turtles and much of this is novel information. However, there seems to be 2 different topics addressed in this paper and that are not completely integrated with one another (information presented here could almost comprise 2 different papers in more wildlife/environmental and/or virus-focused journals, respectively):

A) transcriptome and genome-level profiling of ChHV5 in different types/stages of sea turtle fibropapillomatosis (sample size could be higher but it is hard to get the tissues in endangered species- perhaps with additional samples from other rehabilitation facilities or frozen back samples from other prior studies?).

B) eDNA-based pathogen monitoring showing ChHV5 shed in the water column and correlates with the severity of disease/lesions in sea turtles. This information is particularly interesting and has definite environmental and rehabilitation applications.

Sample size for comparison of viral sequencing reads at the DNA and RNA levels in tumor and non-tumor tissue is very low. Do you have any additional samples available or was it cost-prohibitive?

How is the strength of the host immune response related to viral gene expression (do you have data on inflammation at tumor sites or any complete blood cell counts?)

The could be better focused into subtopics with paragraph subheads related to this study- as there is so much discussed here and there is some lack of consistency and it is hard to follow. I suggest sticking with with one topic per paragraph and the general themes of a paragraph grouped together (e.g. environmental impacts, pathogenesis within the hosts and differences in gene expression).

minor comments:

Introduction- change 'circumglobally' to 'worldwide'

p. 10- change 'largescale' to 'large scale'

Discussion- the paucity of epidermal inclusion bodies are mentioned here for FP-affected animals. However, the authors should also mention the possibility that processes occurring in the dermis (fibroblasts) may also influence the development of tumors- the tumor lesion is not only in the epidermal cells- and viral-host dynamics in the dermis could also be important.

We would like to thank all the reviewers for their careful consideration of the manuscript and for their generous, thoughtful and constructive comments. We have implemented all of the requested changes, and included new data and results, which have helped to greatly improve the manuscript.

Author responses in **red text**.

Reviewers' comments:

Reviewer #1 (Remarks to the Author):

This manuscript clearly reflects the final entering into the fibropapillomatosis genomics and eDNA-based pathogen monitoring. This field has been barely scratched until now, but it can potentially reveal the precise mechanisms through which the virus is transmitted and the role of ChHV5 in host cell transformation and tumor progression. I personally applaud the effort and work done in this study. Such research will provide insights into this wildlife epizootic and reveal how ChHV5 can rapidly induce novel cancer incidence on an epidemic scale. Such information is vital to enable improved management, treatment and mitigation strategies to be developed to combat this sea turtle conservation-relevant disease epizootic.

There are, however, a couple of technical issues and clarifications that I requested to be addressed. Due to the length, complexity and size of the dataset generated in this study, it is completely understandable that more than one publication will be achieved. However, it is also important to check how much overlapping of data and information in general is in between this complementary publication (in press or in review) and the present study. Besides these two big aspects, there are some minor comments, more sort of personal curiosity commentary to the authors (which I would love to chat about it for long periods of time) and suggested changes

We thank the reviewer for their enthusiastic endorsement of the manuscript and its findings. In terms of results there is no overlap between the publications. They share the RNA-seq experiment, but the other publication deals exclusively with the host aspects (reads aligned to green sea turtle genome) of the dataset, while this publication deals exclusively with the viral aspects of the dataset (reads aligned to ChHV5 and PV1 genomes). In addition, each manuscript has its own unique datasets (e.g. eDNA, and tumor qPCR in this manuscript). The sister publication has recently been published in *Communications Biology* and can be accessed at <https://www.nature.com/articles/s42003-021-01656-7>.

We would also be keen to discuss FP in detail with the reviewer and would be more than happy to talk once the review process of the current manuscript has been completed.

We have now implemented all suggested changes in the reviewer's Word document.

Regarding genome coverage, we did not try to assemble a new ChHV5 reference genome. However, it is possible that the reads could be used for this purpose. Our ChHV5 genome coverage (DNA-seq) ranged from 1.7x (non-tumor sample) to 585x (lung tumor sample) coverage per sample. In total there are 38 DNA-seq samples. From the new NovaSeq DNA-seq samples added during the revision alone we have a 2,975x ChHV5 genome coverage. The eDNA tank sequencing sample yielded a ChHV5 genome coverage of 11.6x.

I think if few of the issues that I highlighted are addressed and adjusted, I would definitely give my full recommendation to the Editors to accept this paper, which once more, I enjoyed a lot reading and I will look much forward reading in a final version, once fully accepted.

Reviewer #2 (Remarks to the Author):

This manuscript is chock-full of interesting data. The presentation of the work, however, needs additional editing. For example, there are several parts of the methodology that were either totally left out or abbreviated to the point where it was difficult to understand exactly what was done. This needs to be addressed. There were also some errors in allocation of descriptions to the wrong section- i.e., passages suitable for the discussion section were included in the results, etc. Also, the paper is so dense and multi-faceted to the point of detracting from the most important part, namely the isolation of ChHV5 in tank water samples and the important implications this could have. I recommend restructuring the presentation, adding in more details about the sampling methodology in particular, and highlighting the most relevant and novel parts of the study instead of burying them amongst data that merely supports other, previous studies. Detailed comments are below. Since there were no line numbers, comments are annotated based on sections and page numbers.

Thank you for your comments and thoughtful review. Apologies for any omissions. We have now expanded the methodology descriptions in the manuscript to enable easier understanding of the experiments and sampling. In addition, we have now restructured the entire manuscript, bringing the eDNA results to the fore, as was the consensus between the reviewers.

Title: Is there supposed to be a period at the end of the title?

Removed.

Introduction

p.2:

- northeastern Florida, correct? **Clarified.**
- here and throughout, replace "C. mydas" with "green turtles". **We have retained *C. mydas* in parts of the manuscript, as in some geographic locations (including where FP occurs) green sea turtles are more commonly known as black sea turtles (e.g. some Pacific locations). Therefore, to avoid confusion we opted to also refer frequently to the scientific name of the species.**

- "Long-lived reptiles have normally robust anti-cancer defenses^{24,25}, with the exception of FP, reports of neoplasia in sea turtles are rare." This sentence needs references for reports of non-FP neoplasia in sea turtles. **Added the following references, Orós et al. 2000¹, Orós et al. 2001² and Orós et al 2004³.**

- change "exposures are likely impairing" to: "exposures likely impair..." **Done.**
- remove comma after "While some environmental co-factors" **Done.**

p.3

- "alpha herpesvirus" is just one word **Fixed.**
- "released into the environment and" - Insert comma after "environment" **Done.**

Results

p.4

- "To investigate the hypothesis that ChHV5 might be latent in established tumors, but more active during crucial early stage tumor initiation events, and determine whether ChHV5 was lytic or latent in different tumor types, we sequenced 90 samples (RNA-seq) from five FP tumor types: new growth external (newly formed and unassociated with surgical sites), established external, regrowth external (post-surgical excision), internal lung and internal kidney. The resulting reads were then aligned to the ChHV5 genome⁵⁷." - These details belong in the Methods section. **Moved.**

p.5

- Here and throughout, the statistical methods (Kruskal Wallis, Mann-Whitney, correlation, etc.) described in the Results section were not even mentioned in the Methods section. Please reconcile.

Apologies, this has been amended and the statistical tests are now reported in the Methods also.

- Does 'external FP' differentiate between tumors of the soft skin, eyes, and/or shell? Details on the origins of the tumors used in this study would be really helpful in understanding and interpreting these results.

We have now provided the location of every external tumor in a dedicated column in Supplemental Dataset 1. To help guide the reader towards that data we have included the following sentence to the methods section:

'External tumors were sampled from a range of body locations, including skin, plastron and eye tumors (Supplemental Dataset 1).'

p.7

- "This suggests that rapid proliferation of host tumor cells infected with latent virus is the primary driver of the high viral loads sometimes observed in FP tumors. Fibropapillomatosis tumors can double in size in less than two weeks¹." - This belongs in the Discussion section.

Moved

- at least at first mention, identify the "bladder" as the "urinary bladder" Done.

p.8

- "These results have implications for the spread of ChHV5, as urine is a potential source of ChHV5 transmission^{43,44}." - This belongs in the Discussion section. Moved.

- Depending on the circumstances under which the hatchlings were sampled, this data may be problematic. If the hatchlings were housed in tanks at the rehab facility prior to dying, it is equally (if not more) likely that they were exposed through tank water than being exposed via vertical transmission. If they were housed in tanks prior to death, this should be addressed in the discussion section. If so, I suggest this section be minimized/discussed differently.

The hatchlings were either not housed in tank water prior to death (see below), or housed in a dedicated hatchling tank, which is on a separate water system (sea water drawn directly from the Atlantic) to our FP patients, the hatchling tank is also in its own room in a separate building with no contact between juvenile patients and hatchling patients.

In addition, as part of the new figure panel providing ChHV5 levels in each of the samples pooled for eDNA sequencing, the qPCR-based eDNA results for the hatchling/washback tank confirms that no ChHV5 is present (Supplemental Fig. 2C).

We have also added results showing ChHV5 detection in an unhatched turtle (ChHV5 viral DNA range: 0 - 2.24E-05 pg/ μ l. Amplification ratio: 0.125), which also was never in contact (housed in a different building to both the FP-patients and the hatchling tank) with FP-afflicted patients, or any tank water.

- "If hatchlings are already infected with ChHV5 (vertical transmission), this has serious implications for any potential population-level vaccination-based mitigation strategies. On-beach (immediate nest emergence⁵⁸) sampling and sequencing-based ChHV5 detection should be conducted to confirm this finding and to determine the prevalence of ChHV5 infected hatchlings." - This belongs in the Discussion section. **Moved.**

- The 2nd paragraph prior to the sentence starting with "Therefore," belongs in the Discussion section. **Moved.**

p.9

- This is by far the strongest finding of this study. I recommend that the manuscript be structured to lead with this result. **Thank you for the suggestion. We have now restructured the manuscript, so that it leads with the eDNA findings first.**

- This is the first mention that the leeches were sourced from FP tumors. This methodology should be described in detail in the methods section. **Now described in detail in the methods section.**

- What statistical test was used to analyze this correlation? **Details now added to the methods section. A Pearson Correlation Coefficient test was used.**

p.11

- As stated in my above comment, if the hatchlings were housed in tank water prior to death, these results are likely overstated and should be toned down. **The hatchlings were either not housed in tank water prior to death, or housed in a dedicated hatchling tank (please see the response to p.8 point above), which is on a separate water system (sea water drawn directly from the Atlantic) to our FP patients, the hatchling tank is also in its own room in a separate building with no contact between juvenile patients and hatchling patients.**

Given the potential implications of this finding we have also now included ChHV5 qPCR detection results from a pre-hatchling, which never left its egg before dying. This pre-hatchling was part of a set of 3 eggs brought to the hospital (after a nest evaluation, the remaining 3 unhatched eggs after the remainder of the clutch had already emerged from the nest). The other two clutch mates successfully hatched, with only one individual dying. These eggs were housed in a separate building to both our juvenile and hatchling patients, and were also never placed in any water. A number of tissue samples from this deceased pre-hatchling tested positive for ChHV5 at low viral loads.

p.12

- What about regular or intermittent monitoring of tank water in captive turtle populations, as a disease prevention strategy? I think this warrants mentioning here. **This has now been included.**

Thank you for the suggestion, we fully agree, indeed we currently have a separate manuscript in press which reviews aquatic eDNA's potential benefits to disease detection and prevention.

p.13

- Which non-tumor areas, specifically? Soft skin? From which body part(s)? Be more specific.

We have now included the following sentence: 'External tumors were sampled from a range of body locations, including skin, plastron and eye tumors (Supplemental Dataset 1)'. For completeness, the external location of each sample (tumor and non-tumor) has now been added to Supplemental Dataset 1 (previously named Supplemental Table 2 in the initial submission).

- What type(s) of non-tumor tissues were sampled? I.e., what organ system(s)? Be more specific.

Apologies, we have now added the underlined section to highlight the internal tissue types sampled from deceased juvenile turtles during necropsy: 'Internal tissue samples (tumors and non-tumor tissue samples from lung, kidney, bladder, liver, mouth, sub-cutaneous and heart) were..'.
'

- You may want to point out that the euthanasia of turtles with internal tumors is based on a protocol recommended by FWC. **Thank you, added.**

- What about the hatchling turtle samples? Sampling of hatchlings needs to be described in detail here. **Added.**

- Up to how many weeks/months were samples stored prior to DNA extraction? **The storage durations for each sample type have now been added to the Methods section. For these particular samples this was a range, with samples being stored between <1 day (immediate extraction, hours after being added to RNAlater) and 8 months, with 85% of samples being extracted within one month.**

p.15

- How were these hatchlings encountered/housed? Were they kept in tanks prior to dying? Or were they found dead in the nest or on the beach? If kept in tanks, were they exposed to tank water shared by juvenile green turtles in rehab? **Hatchlings are brought to the hospital by nest patrol teams. After nest emergence, nests are assessed and any remaining hatchlings not strong enough for immediate release are brought to the hospital. Please see response above re p11 comment, in relation to their complete separation of juvenile and FP positive patients.**

- As written, this sounds like only a single tank sample was taken. The tank water sampling protocol needs to be described in more detail here. **Changed to 'pooled library' and 'pooled sample' and added further details, as requested. We have also now added ChHV5 qPCR results**

for each of the five individual eDNA extractions (Tank 1-5) which were pooled for sequencing (Supplemental Fig. 2c). At the time a portion of each extraction was pooled to generate the single tank eDNA sequencing library, with the remainder of the extracted DNA being saved for subsequent qPCR-based analysis.

- Why were these samples combined from different tanks? This needs to be explained and justified. Or maybe I am just confused by the way this is written. Either way, please clarify. We have now tried to better clarify this point in the manuscript's text (including the addition of Supplemental Fig. 2c). For all qPCR-assessed eDNA samples, samples were never pooled, but investigated individually. However, for WGS we pooled five samples, as it was cost prohibitive to sequence each sample individually. Since we only had funding for a single tank eDNA WGS, we pooled multiple samples to provide an average ChHV5 RPTM value for comparison with the tissue samples. This was done to avoid having eDNA WGS data from only a single data point which would be more prone to outliers. Samples were pooled in equal volume to obtain an average of ChHV5 reads across the 5 tanks.

Subsequently, each of these 5 tank eDNA samples (the portions, not pooled and used for WGS) were also assessed individually for ChHV5 by qPCR. To provide a more comprehensive picture of these five samples, this data has now been added to the supplemental material (Supplemental Fig. 2c), as predicted there was a range of ChHV5 quantity across the samples.

The loggerhead tank was also included in the pool, as (to be cost effective) the eDNA WGS sample was also used for a separate study relating to sea turtle species specific eDNA detection from water samples (turtle aligning portions of the eDNA sequencing results).

p.17

- What types of tissue samples? This is unclear. Sampling details on what tissue types, from what turtles, are needed. We have now added a new supplemental table detailing the origins of all samples used for the tissue qPCR (Supplemental Dataset 3).

- There is actually no description of the sampling of the leeches. Where were they removed from? From turtles with FP? Where on the turtles' bodies were they attached? Were the leeches ground up whole and DNA extracted from that? Or were they dissected and the guts subjected to extraction? Apologies for this oversight, additional details regarding the leech sampling have now been added to the 'Tissue sampling' section of the methods. In addition, we have added additional leech results (Fig. 1a,d,e and Supplemental Table 1).

- As noted above, details should be clearly explained in the tank water sampling methods section, above. As requested, additional tank sampling details have now been included.

p.18

- Was total tumor surface area compared to turtle size for a % of body affected? A brief description of this methodology is warranted. **As requested, we have now added a brief description of the methodology.**

No, non-tumor turtle surface area was not calculated as we were interested in the relationship between overall tumor size and ChHV5 shedding.

Since FP tumors regularly grow over unaffected areas (connected by a 'stalk' but not actually affecting all areas 'under' the tumor), it is not a straightforward task to measure unaffected external turtle surfaces versus tumor surface area. Every tumor would need to be physically manipulated to calculate the area of unaffected skin/shell obscured by the tumor growth. Our FWC research permit only allows us to image patients and measure tumors with calipers, we are not permitted for prolonged measurements of non-tumor area. For comparisons between patients, we more commonly use pre-existing rehabilitation data such as turtle weight, or tumor burden score.

- Were these slides reviewed by a veterinary pathologist? There should be a complete description of how the slides were reviewed. **Yes, the slides were reviewed and scored by a professional veterinary pathologist (Brian Stacy, NOAA & UF). As requested, complete description of this review has now been added to the methods section.**

Table 1

- Do these genes have associated GenBank accession numbers? If so, it would be good to include those in this table for reference.

All gene sequences are from the ChHV5 reference genome. As such we have added the following sentence to the Table legend: 'Gene names correspond to the ChHV5 reference genome [GenBank accession number: HQ878327.2].'

Table 3

- Not sure this table is necessary, or maybe just include it as a supplement. It should suffice to summarize in the text that 0 reads were detected for all but one sample.

Table 3 has been moved to the supplemental as suggested (now Supplemental Table 3). It now also has data for an additional 18 DNA-seq libraries. ChHV5 results from these new 18 samples have also been added to Fig. 5a.

Figure 1B. This is interesting, in that it appears to somewhat contradict what you found with tank water, in that patients with large well-established tumors shed more virus into tank water than those with small new-growth tumors. If real, I think this contradiction should be addressed in the discussion.

We agree that the predominant latency across tumor types is interesting. However, we do not believe that these findings are necessarily contradictory. While there is no significant difference between the amount of ChHV5 RNA detected in established and new growth tumors (across similar sized tumor sub-samples), as tumors become larger and well-established, they would be expected to cumulatively produce more virus, even if the level of virus production per portion of tumor is comparable to that of new growth tumors. In other words, if the ChHV5 production rate was constant per mm³ of tumor, having a larger tumor burden would result in more virus being produced and shed into the water column (without requiring the rate of ChHV5 production per mm³ of tissue to increase).

Tank water is a cumulative readout of shedding from all tumors simultaneously, whereas tissue sampling only provides a read out of a portion of an individual tumor.

Figure 2B. These data seem to suggest that the mechanisms driving internal tumor development are different from those driving external tumor development. This is touched on in the discussion but could be addressed more robustly, perhaps with examples/comparisons to better-known herpesviruses such as Kaposi's sarcoma.

We agree with the reviewer's assessment regarding external and internal tumors. Interestingly, we also found differences in the host (green turtle) genes driving internal and external FP tumors (which is discussed in the sister host paper to this virally focused manuscript, <https://doi.org/10.1038/s42003-021-01656-7>). As suggested, we have now added additional text regarding this to the discussion, including additional references to human oncogenic viruses:

'The differing ChHV5 gene expression profiles between internal and external FP tumors, suggests that the mechanisms driving internal tumor development are different from those driving external tumor development. Supporting this, we recently showed that the host oncogenic signaling events driving internal and external tumor development also differ dramatically (Yetsko et al. 2021). Even while in their latent stage oncogenic viruses can manipulate host cell signaling and contribute to oncogenic transformation and tumor development⁴⁻⁶. Latency is also used as an immune avoidance strategy, with bouts of temporally and spatially restricted lytic activity helping to prevent a largescale immune response. This feature of oncogenic viruses may help to explain why ChHV5 is predominantly latent in FP tumors, and why lytic viral replication is so spatially restricted^{7,8}. In this context, FP tumors with higher ChHV5 expression levels may be more prone to inducing immune responses, leading to improved patient outcomes. Further supporting this hypothesis, we recently showed that, in the same patient cohort used in this study, higher expression levels of immune-related host genes was associated with better patient outcomes (Yetsko et al. 2021).'

Figure 3A. Why do the authors think this lung sample had such a high number of reads? Any

clues about why this would be, based on known herpesvirus biological mechanisms? Please address in the discussion.

There is a wide range of ChHV5 (DNA) load across the lung tumor samples we assessed by qPCR (Figure 3C, figure order from the original submission). Therefore, that the one lung tumor assessed by DNA-seq had such a high level of ChHV5 DNA (3,673 RPTM) may not be representative of all lung tumors, merely be the most extreme range of normal inter-tumor ChHV5 load variability we report. We have now added DNA-seq results for an additional 18 libraries to the manuscript. With the addition of these new samples, while the lung tumor still has the highest load, it is not so far above the next highest load samples (DNA-seq), a kidney tumor with 3,127 RPTM and an external tumor with 3,031 RPTM. Based on the qPCR results, and the new DNA-seq results we do not believe that this lung tumor difference is now sufficient to merit special mention in the Discussion.

However, we were originally tempted to hypothesize that the lung may have a higher load due to the tissue/air interface in the lung. ChHV5 was only successfully cultured in cells (after decades of attempts) when the cells had a feeder cell layer and were exposed to air (not fully submerged in media). Herpesviruses are known to preferentially switch to lytic replication at the skin surface (assumed to be an adaptive transmission mechanism, i.e., escape from the body), so this may be behind the requirement for a tissue/air interface. However, that same lung tumor with the high ChHV5 DNA load (3,673 RPTM) had negligible ChHV5 RNA load (68 RPTM), meaning it was most likely latent in the lung tumor also.

If even given the additional DNA-seq results you feel that lung tumor still warrants special mention in the Discussion we would be happy to do so.

Figure 4A. This is mostly skimmed over in the manuscript, probably because it was not well described in the methods section. Please reconcile and describe in more detail if you are going to include it at all.

Apologies, the full details have now been included in the methods section. In addition, we have expanded the discussion of the result, and added an additional figure panels with further leech ChHV5 detection and ChHV5 detection from sand eDNA (Figs 1 and 2).

Reviewer #3 (Remarks to the Author):

General comments:

This paper provides useful genomic information and environmental information about fibropapillomatosis in sea turtles and much of this is novel information. However, there seems to

be 2 different topics addressed in this paper and that are not completely integrated with one another (information presented here could almost comprise 2 different papers in more wildlife/environmental and/or virus-focused journals, respectively):

A) transcriptome and genome-level profiling of ChHV5 in different types/stages of sea turtle fibropapillomatosis (sample size could be higher but it is hard to get the tissues in endangered species- perhaps with additional samples from other rehabilitation facilities or frozen back samples from other prior studies?).

B) eDNA-based pathogen monitoring showing ChHV5 shed in the water column and correlates with the severity of disease/lesions in sea turtles. This information is particularly interesting and has definite environmental and rehabilitation applications.

Thank you for your comments and thoughtful review. While we understand the Reviewer's assessment regarding potentially splitting the paper, we also feel it is important to have both datasets presented together as they provide highly inter-related data. We believe that the two data types warrant inclusion in a single manuscript, for the reasons outlined below. However, we have taken the Reviewer's comment onboard and have tried to more explicitly highlight this inter-dependence of the environmental and tissue data types in the revised manuscript (including a new summary table, Table 3).

The level of virus being shed into the water column must have some relationship to the viral activity within the host/tumor. From the eDNA data we show that more ChHV5 DNA is present in the water column for patients with higher tumor burdens. While internally, there is no significant difference between the level of viral gene expression between established and new growth tumors, with the virus remaining predominantly latent in all tumor types (generally high ChHV5 DNA levels, but low ChHV5 RNA levels, with expression being largely restricted to latency-associated genes). Therefore, combining the eDNA and genomic data leads to an important conclusion, that a higher tumor burden leads to more viral shedding, but that this is driven primarily by larger tumors producing a similar level of virus per area as new growth tumors, rather than established tumors producing more lytic virus and more shedding (per area of tumor). I.e. there is no significant shift towards lytic virus production in established tumors, rather levels of lytic virus remain relatively low, but the sheer increase in tumor volumes results in more shedding. This is an important distinction, given the 'super-spreader' hypothesis previously postulated for FP transmission.

The reviewer is correct, that it can be difficult to generate large FP sample sizes. However, the main limiting factor in this study is that it is highly cost-prohibitive to have large sequencing sample sizes (as we sequence not just the small viral genome/transcriptome, but also the much larger host genome/transcriptome for each sample). It costs approximately \$650 USD per RNA-seq sample and approximately \$1,500 per DNA-seq sample.

That being said, we have now included an additional 18 additional DNA-seq samples, including samples from other facilities in this revised version of the manuscript.

Total numbers of samples assessed by each technology in the resubmitted manuscript:

- 90 samples assessed by whole transcriptome RNA-seq,
- 38 samples assessed by whole genome DNA-seq,
- 79 samples (host tissue only, not including eDNA, or leeches) assessed by targeted qPCR.

This is by far the largest number of whole genome and whole transcriptome ChHV5 samples yet sequenced.

While our sample numbers may be small compared to some human and lab animal studies, they are large for the current standards of the wildlife and sea turtle FP genomics fields. The previous largest WGS (viral, no host genomes) of ChHV5 DNA was 8 samples⁹, the previous ChHV5 whole transcriptome was our 2018 paper which had 10 samples¹⁰ (viral and host) and a new pre-print with 50 whole transcriptome sequences¹¹ (viral and host).

Sample size for comparison of viral sequencing reads at the DNA and RNA levels in tumor and non-tumor tissue is very low. Do you have any additional samples available or was it cost-prohibitive?

Again, the reviewer is correct, it is cost prohibitive (see response to the point above), however we have now added results for an additional 18 whole genome sequencing samples.

We have an established biobank of patient matched FP and non-tumor samples, the samples in this manuscript are the maximum number we could afford to sequence currently.

How is the strength of the host immune response related to viral gene expression (do you have data on inflammation at tumor sites or any complete blood cell counts?)

An excellent question. We have profiled the host immune response in a sister paper to this viral manuscript. The sister paper examined the host gene expression profiling aspects of the RNA-seq data used for ChHV5 analysis in this viral manuscript. The host paper has recently been published by *Communications Biology*¹². We have now added some additional details on the relationship between viral expression and host response to the Discussion section of the resubmitted manuscript. Briefly, higher ChHV5 viral gene expression was associated with better patient outcomes (current manuscript), while high levels of immune-related genes were associated with good patient outcomes (sister host RNA-seq paper). Therefore, we postulate that the more lytic virus is present in tumors, the higher the host immune response, leading to improved neutralization of the virus and tumor cells. Other oncogenic viruses are known to remain predominantly latent to help evade host immune systems.

The could be better focused into subtopics with paragraph subheads related to this study- as there is so much discussed here and there is some lack of consistency and it is hard to follow. I suggest sticking with with one topic per paragraph and the general themes of a paragraph grouped together (e.g. environmental impacts, pathogenesis within the hosts and differences in gene expression).

We have revised the discussion section in line with this comment. However, we did not add sub-headings as that is not allowable within the journal's style guidelines. If the Editor is willing to allow Discussion sub-headings we would be happy to incorporate them. We have also added a new summary table (Table 3) to help highlight the main points and the inter-relatedness of the findings.

minor comments:

Introduction- change 'circumglobally' to 'worldwide' **Changed.**

p. 10- change 'largescale' to 'large scale' **Changed.**

Discussion- the paucity of epidermal inclusion bodies are mentioned here for FP-affected animals. However, the authors should also mention the possibility that processes occurring in the dermis (fibroblasts) may also influence the development of tumors- the tumor lesion is not only in the epidermal cells- and viral-host dynamics in the dermis could also be important.

We agree with the reviewer and have now added the following sentence to the discussion: 'Processes occurring in the dermis (especially relating to fibroblasts) may also influence the development of tumors and host-viral interactions, as FP tumors involve changes to both epidermal and dermal cells.'

References:

- 1 Orós, J. & Torrent, A. Unusual tumors in three loggerhead sea turtles (*Caretta caretta*) stranded in the Canary Islands, Spain. *Mar. Turtle Newsl.* **88**, 6-7 (2000).
- 2 Orós, J. *et al.* Multicentric lymphoblastic lymphoma in a loggerhead sea turtle (*Caretta caretta*). *Vet. Pathol.* **38**, 464-467 (2001).
- 3 Orós, J., Tucker, S., Fernández, L. & Jacobson, E. R. Metastatic squamous cell carcinoma in two loggerhead sea turtles *Caretta caretta*. *Dis. Aquat. Org.* **58**, 245-250 (2004).
- 4 Young, L. S. & Murray, P. G. Epstein–Barr virus and oncogenesis: from latent genes to tumours. *Oncogene* **22**, 5108-5121, doi:10.1038/sj.onc.1206556 (2003).
- 5 Ye, F., Lei, X. & Gao, S.-J. Mechanisms of Kaposi's Sarcoma-Associated Herpesvirus Latency and Reactivation. *Advances in Virology* **2011**, 193860, doi:10.1155/2011/193860 (2011).

- 6 Young, L. S., Yap, L. F. & Murray, P. G. Epstein–Barr virus: more than 50 years old and still providing surprises. *Nature Reviews Cancer* **16**, 789-802, doi:10.1038/nrc.2016.92 (2016).
- 7 Work, T. M., Dagenais, J., Balazs, G. H., Schettle, N. & Ackermann, M. Dynamics of Virus Shedding and In Situ Confirmation of Chelonid Herpesvirus 5 in Hawaiian Green Turtles With Fibropapillomatosis. *Vet. Pathol.* **52**, 1195-1201, doi:10.1177/0300985814560236 (2015).
- 8 Kang, K. I. *et al.* Localization of Fibropapilloma-associated Turtle Herpesvirus in Green Turtles (*Chelonia mydas*) by In-Situ Hybridization. *Journal of Comparative Pathology* **139**, 218-225, doi:<http://dx.doi.org/10.1016/j.jcpa.2008.07.003> (2008).
- 9 Morrison, C. L. *et al.* Genomic evolution, recombination, and inter-strain diversity of chelonid alphaherpesvirus 5 from Florida and Hawaii green sea turtles with fibropapillomatosis. *PeerJ* **6**, e4386, doi:10.7717/peerj.4386 (2018).
- 10 Duffy, D. J. *et al.* Sea turtle fibropapilloma tumors share genomic drivers and therapeutic vulnerabilities with human cancers. *Communications Biology* **1**, 63, doi:10.1038/s42003-018-0059-x (2018).
- 11 Blackburn, N. B. *et al.* Transcriptomic profiling of fibropapillomatosis in green sea turtles (*Chelonia mydas*) from South Texas. *bioRxiv*, 2020.2010.2029.360834, doi:10.1101/2020.10.29.360834 (2020).
- 12 Yetsko, K. *et al.* Molecular characterization of a marine turtle tumor epizootic, profiling external, internal and postsurgical regrowth tumors. *Communications Biology* **4**, 152, doi:10.1038/s42003-021-01656-7 (2021).

REVIEWERS' COMMENTS:

Reviewer #2 (Remarks to the Author):

The authors have done a fine job of revising the manuscript. It is much clearer now– easy to read and very informative. Great job! I have only a few additional, minor suggested edits, detailed below.

L77: insert “often” in front of “includes”

L400–407: Need to point out somewhere in this paragraph that leeches likely represent a mechanical vector, as studies to determine whether leeches function as a true vector of ChHV5 (i.e., we don’t know whether leeches harbor viral replication).

L427: change “form” to “from”

L481–490: I still have questions about the way the hatchlings were sampled. Were the skin/tissue samples collected using sterile technique? Were the hatchlings necropsied all at the same time, perhaps along with other turtles as well? These factors play into potential sources of contamination at the point of sampling. I belabor this point because finding ChHV5 in hatchlings contradicts several previous studies that point to direct horizontal transmission occurring at neritic foraging grounds– and I believe many readers will have similar questions.

L500: typo – “The” has an extra “M” in it

Reviewer #3 (Remarks to the Author):

All reviewers comments have been adequately addressed, in my opinion, and I appreciate the table of key findings that summarizes all of the important findings. This article advances the field of wildlife diseases, particularly in our understanding of the transmission biology of fibropapillomatosis in sea turtles, and raises many important questions regarding the immunopathology and epizootiology of this critically important disease.

Reviewer #2 (Remarks to the Author):

The authors have done a fine job of revising the manuscript. It is much clearer now– easy to read and very informative. Great job! I have only a few additional, minor suggested edits, detailed below.

Thank you for these supportive comments, and for the time you have taken to review the manuscript.

L77: insert “often” in front of “includes” **Inserted.**

L400–407: Need to point out somewhere in this paragraph that leeches likely represent a mechanical vector, as studies to determine whether leeches function as a true vector of ChHV5 (i.e., we don’t know whether leeches harbor viral replication). **We agree that this is an important clarification (especially as leeches without prominent blood pellets tested negative for ChHV5) and have now inserted the following sentence: ‘Leeches likely only represent a mechanical vector of ChHV5, as studies to determine whether leeches function as a true vector with ChHV5 replicating within leeches (as an intermediary host) have not yet been conducted.’**

L427: change “form” to “from” **fixed**

L481–490: I still have questions about the way the hatchlings were sampled. Were the skin/tissue samples collected using sterile technique? Were the hatchlings necropsied all at the same time, perhaps along with other turtles as well? These factors play into potential sources of contamination at the point of sampling. I belabor this point because finding ChHV5 in hatchlings contradicts several previous studies that point to direct horizontal transmission occurring at neritic foraging grounds– and I believe many readers will have similar questions. **Yes, sterile technique was used.**

No, they were sampled separately. As a further precaution the hatchling sampling was also conducted in separate lab, with separate instrumentation (single use disposable) to any other turtles.

We appreciate the novelty of these findings and the reviewer’s query, therefore we have added the following sentence to the methods section: ‘This sampling was conducted using sterile technique, with separate instrumentation (single use disposable), and was conducted in a lab which never housed FP-patients or any patients from other size classes.’

Further supporting the absence of contamination, the ChHV5 genome sequence from the leatherback hatchling, does not cluster with the ChHV5 genomes recovered from any of our sequenced FP tumor samples (unpublished data, analysis conducted as part of an ongoing ChHV5 phylogenomic project). The leatherback hatchling sample was the only hatchling sample we assessed in that phylogenomic study, as it was the only one above the 10x genome coverage

cut-off employed in that study. Rather, the ChHV5 genome from the leatherback clustered more closely to ChHV5 genomes obtained from Kemp's ridley turtles. The Kemp's samples were not processed or sequenced at our lab, meaning no contamination with such ChHV5 strains could have occurred.

L500: typo – "The" has an extra "M" in it **fixed**

Reviewer #3 (Remarks to the Author):

All reviewers' comments have been adequately addressed, in my opinion, and I appreciate the table of key findings that summarizes all of the important findings. This article advances the field of wildlife diseases, particularly in our understanding of the transmission biology of fibropapillomatosis in sea turtles, and raises many important questions regarding the immunopathology and epizootiology of this critically important disease.

Thank you for your supportive comments, and for the time you have taken to review the manuscript.